# DEEP JUMP Q-EVALUATION FOR OFFLINE POLICY EVALUATION IN CONTINUOUS ACTION SPACE

## ABSTRACT

We consider off-policy evaluation (OPE) in continuous action domains, such as dynamic pricing and personalized dose finding. In OPE, one aims to learn the value under a new policy using historical data generated by a different behavior policy. Most existing works on OPE focus on discrete action domains. To handle continuous action space, we develop a brand-new deep jump Q-evaluation method for OPE. The key ingredient of our method lies in adaptively discretizing the action space using deep jump Q-learning. This allows us to apply existing OPE methods in discrete domains to handle continuous actions. Our method is further justified by theoretical results, synthetic and real datasets.

## 1 INTRODUCTION

Individualization proposes to leverage omni-channel data to meet individual needs. Individualized decision making plays a vital role in a wide variety of applications. Examples include customized pricing strategy in economics (Qiang & Bayati, 2016; Turvey, 2017), individualized treatment regime in medicine (Chakraborty, 2013; Collins & Varmus, 2015), personalized recommendation system in marketing (McInerney et al., 2018; Fong et al., 2018), etc. Prior to adopting any decision rule in practice, it is crucial to know the impact of implementing such a policy. In many applications, it is risky to run a policy online to estimate its value (see, e.g., Li et al., 2011). Off-policy evaluation (OPE) thus attracts a lot of attention by learning the policy value offline using logged historical data.

Despite the popularity of developing OPE methods with a finite set of actions (see e.g., Dudík et al., 2011; 2014; Swaminathan et al., 2017; Wang et al., 2017), less attention has been paid to continuous action domains, such as dynamic pricing (den Boer & Keskin, 2020) and personalized dose finding (Chen et al., 2016). Recently, a few OPE methods have been proposed to handle continuous actions (Kallus & Zhou, 2018; Sondhi et al., 2020; Colangelo & Lee, 2020). All these methods rely on the use of a kernel function to extend the inverse probability weighting (IPW) or doubly robust (DR) approaches developed in discrete action domains. They suffer from three limitations. First, the validity of these methods requires the conditional mean of the reward given the feature-action pair to be a smooth function over the action space. This assumption could be violated in applications such as dynamic pricing, where the expected demand for a product has jump discontinuities as a function of the charged price (den Boer & Keskin, 2020). Second, the value estimator could be sensitive to the choice of the bandwidth parameter in the kernel function. It remains challenging to select this hyperparameter. Kallus & Zhou (2018) proposed to tune this parameter by minimizing the mean squared error of the resulting value estimator. However, their method is extremely computationally intensive in moderate or high-dimensional feature space; see Section 5 for details. Third, these kernel-based methods typically use a single bandwidth parameter. This is sub-optimal in cases where the second-order derivative of the conditional mean function has an abrupt change in the action space; see the toy example in Section 3.1 for details.

To address these limitations, we develop a deep jump Q-evaluation (DJQE) method by integrating multi-scale change point detection (see e.g., Fryzlewicz, 2014), deep learning (LeCun et al., 2015) and OPE in discrete action domains. The key ingredient of our method lies in adaptively discretizing the action space using deep jump Q-learning. This allows us to apply IPW or DR methods to handle continuous actions. It is worth mentioning that our method does not require kernel bandwidth selection. Theoretically, we show it allows the conditional mean to be either a continuous or piecewise function of the action (Theorems 1 and 2) and converges faster than kernel-based OPE (Theorem 3). Empirically, we show it outperforms state-of-the-art OPE methods in synthetic and real datasets.

## 2 PRELIMINARIES

We first formulate the OPE problem. We next discuss the kernel-based OPE methods and multi-scale change point detection, since our proposal is closely related to them.

### 2.1 OFF-POLICY EVALUATION

The observed datasets can be summarized into $\{(X_i, A_i, Y_i)\}_{1 \leq i \leq n}$ where $O_i = (X_i, A_i, Y_i)$ denotes the feature-action-reward triplet for the $i$th subject and $n$ denotes the total sample size. We assume these data triplets are independent copies of some population variables $(X, A, Y)$. Let $\mathcal{X}$ and $\mathcal{A}$ denote the feature and action space, respectively. We focus on the setting where $\mathcal{A}$ is one-dimensional, as in dynamic pricing and personalized dose finding. A deterministic policy $\pi : \mathcal{X} \to \mathcal{A}$ determines the action to be assigned given the observed feature. We use $b$ to denote the behavior policy that generates the observed data. Specifically, $b(\bullet|x)$ denotes the probability density or mass function of $A$ given $X = x$, depending on whether $A$ is continuous or not. Define the expected reward function conditional on the feature-action pair as

$$Q(x, a) = \mathbb{E}\{Y|X = x, A = a\}.$$

We refer to this function as the Q-function, to be consistent with the literature on developing individualized treatment regime (Murphy, 2003).

As standard in the OPE and the causal inference literature (see e.g., Chen et al., 2016), we assume the stable unit treatment value assumption (SUTVA), no unmeasured confounders assumption, and the positivity assumption are satisfied. These assumptions guarantee that a policy's value is estimable from the observed data. Specifically, for a given target policy $\pi$, its value can be represented by

$$V(\pi) = \mathbb{E}\{Q(X, \pi(X))\}.$$

The goal of the OPE is to learn the value $V(\pi)$ based on the observed data.

### 2.2 KERNEL-BASED OPE

For discrete action, Zhang et al. (2012) and Dudík et al. (2011) proposed a DR estimator of $V(\pi)$ by

$$\frac{1}{n}\sum_{i=1}^{n} \psi(O_i, \pi, \widehat{Q}, \widehat{b}) = \frac{1}{n}\sum_{i=1}^{n}\left[\widehat{Q}(X_i, \pi(X_i)) + \frac{\mathbb{I}(A_i = \pi(X_i))}{\widehat{b}(A_i|X_i)}\{Y_i - \widehat{Q}(X_i, \pi(X_i))\}\right], \quad (1)$$

where $\mathbb{I}$ denotes the indicator function, $\widehat{Q}$ and $\widehat{b}$ denote some estimators for the Q-function and the behavior policy. The second term $\widehat{b}^{-1}(A_i|X_i)\mathbb{I}(A_i = \pi(X_i))\{Y_i - \widehat{Q}(X_i, \pi(X_i))\}$ inside the bracket corresponds to an augmentation term. Its expectation equals zero when $\widehat{Q} = Q$. The purpose of adding this term is to offer additional protection against potential model misspecification of the Q-function. Such an estimator is doubly-robust in the sense that its consistency relies on either $\widehat{Q}$ or $\widehat{b}$ to be correctly specified. By setting $\widehat{Q} = 0$, equation 1 is reduced to the IPW estimator.

In continuous action domains, the indicator function $\mathbb{I}(A_i = \pi(X_i))$ equals zero almost surely. Consequently, naively applying equation 1 yields the plug-in estimator $\sum_{i=1}^{n} \widehat{Q}(X_i, \pi(X_i))/n$. To address this concern, the kernel-based OPE proposed to replace the indicator function in equation 1 with a kernel function $K\{(A_i - \pi(X_i))/h\}$ with some bandwidth parameter $h$, i.e.,

$$\frac{1}{n}\sum_{i=1}^{n} \psi_h(O_i, \pi, \widehat{Q}, \widehat{b}) = \frac{1}{n}\sum_{i=1}^{n}\left[\widehat{Q}(X_i, \pi(X_i)) + \frac{K\{(A_i - \pi(X_i))/h\}}{\widehat{b}(A_i|X_i)}\{Y_i - \widehat{Q}(X_i, \pi(X_i))\}\right].$$

The bandwidth $h$ represents a trade-off. The variance of the resulting value estimator decays with $h$. Yet, its bias increases with $h$. More specifically, it follows from Theorem 1 of Kallus & Zhou (2018) that the leading term of the bias is equal to

$$h^2 \frac{\int u^2 K(u) du}{2} \mathbb{E}\left(\left.\frac{\partial^2 Q(X, a)}{\partial a^2}\right|_{a=\pi(X)}\right). \quad (2)$$

To ensure the term in 2 decays to zero as $h$ goes to 0, it requires the expected second derivative of the Q-function to exist, and thus $Q(x, a)$ needs to be a smooth function of $a$. However, as commented in the introduction, this assumption could be violated in applications such as dynamic pricing.

Table 1: The bias and the standard deviation (in parentheses) of the estimated values for $V^{(1)}$ and $V^{(2)}$, using DJQE and kernel-based methods. $n = 100$, $X, A \sim \text{Unif}[0,1]$, $Y|X, A \sim N\{Q(X,A), 1\}$. The target policy is given by $\pi(x) = x$.

| Method | DJQE | Kernel (small $h$, $h = 0.8$) | Kernel (large $h$, $h = 2$) |
|---|---|---|---|
| $V^{(1)}(\pi)$ | 0.34 (0.13) | 0.65 (0.17) | 0.33 (0.09) |
| $V^{(2)}(\pi)$ | 0.11 (0.33) | 0.03 (0.32) | 1.21 (0.12) |

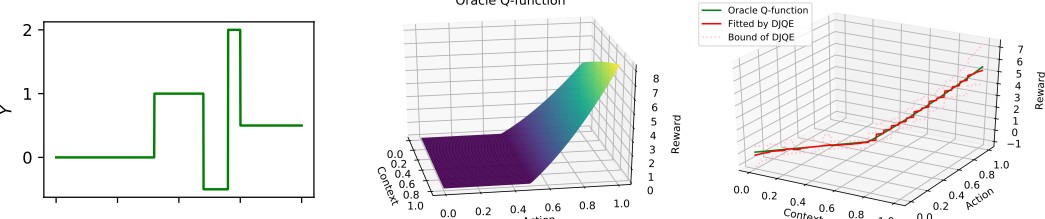

Figure 1: Left panel: example of piece-wise constant function in change point literature. Middle panel: the oracle Q-function on the feature-action space. Right Panel: the green curve presents the oracle Q-function $Q(x, \pi(x))$ under policy $\pi(x) = x$; and the red curve is the fitted mean value by DJQE and the pink dash line corresponds to the 95% confidence bound.

### 2.3 Multi-Scale Change Point Detection

The change point analysis considers an ordered sequence of data, $Y_{1:n} = \{Y_1, \cdots, Y_n\}$, with unknown change point locations, $\tau = \{\tau_1, \cdots, \tau_K\}$ for some unknown integer $K$. Here, $\tau_i$ is an integer between 1 and $n - 1$ inclusive, and satisfies $\tau_i < \tau_j$ for $i < j$. These change points split the data into $K+1$ segment. Within each segments, the expected response is a constant function (see the left panel of Figure 1 for details). A number of methods have been proposed on estimating change points (see for example, Boysen et al., 2009; Killick et al., 2012; Frick et al., 2014; Fryzlewicz, 2014, and the references therein), by minimizing a penalized objective function:

$$\arg\min_{\tau, K} \left( \frac{1}{n} \sum_{i=1}^{K+1} \left[ \mathcal{C}\{Y_{(\tau_{i-1}+1):\tau_i}\} \right] + \gamma K \right),$$

where $\mathcal{C}$ is a cost function that measures the goodness-of-the-fit of the constant function within each segment and $\gamma K$ penalizes the number of change points with some regularization parameter $\gamma$. We remark that all the above cited works focused on models without features. Our proposal goes beyond these works in that we consider models with features and use deep neural networks (DNN) to capture the complex relationship between the response and features.

## 3 Deep Jump Q-Evaluation

In section 3.1, we use a toy example to demonstrate the limitation of kernel-based methods. We present the main idea of our algorithm in Section 3.2. Details are given in Section 3.3.

### 3.1 Toy Example

As discussed in the introduction, existing kernel-based OPE methods use a single bandwidth to construct the value estimator. Ideally, the bandwidth $h$ in the kernel $K\{(A_i - \pi(X_i))/h\}$ shall vary with $\pi(X_i)$ to improve the accuracy of the value estimator. To elaborate this, consider the Q-function $Q(x, a) = 10 \max\{a^2 - 0.25, 0\} \log(x + 2)$ for any $x, a \in [0, 1]$. By definition, the Q-function is smooth over the entire feature-action space. However, it has different "patterns" when the action belongs to different intervals. Specifically, for $a \in [0, 0.5]$, $Q(x, a)$ is constant as a function of $a$. For $a \in (0.5, 1]$, $Q(x, a)$ depends quadratically in $a$. See the middle panel of Figure 1 for details.

Consider the target policy $\pi(x) = x$. We decompose the value $V(\pi)$ into $V^{(1)}(\pi) + V^{(2)}(\pi)$ where

$$V^{(1)}(\pi) = \mathbb{E}Q(X, \pi(X))\mathbb{I}(\pi(X) \leq 0.5) \text{ and } V^{(2)}(\pi) = \mathbb{E}Q(X, \pi(X))\mathbb{I}(\pi(X) > 0.5).$$

Similarly, denote the corresponding kernel-based value estimators by

$$\widehat{V}_h^{(1)}(\pi) = \frac{1}{n}\sum_{i=1}^{n}\psi_h(O_i,\pi,\widehat{Q},\widehat{b})\mathbb{I}(\pi(X_i)\leq 0.5) \text{ and } \widehat{V}_h^{(2)}(\pi) = \frac{1}{n}\sum_{i=1}^{n}\psi_h(O_i,\pi,\widehat{Q},\widehat{b})\mathbb{I}(\pi(X_i) > 0.5).$$

Since $Q(x,a)$ is a constant function of $a \in [0, 0.5]$, its second-order derivative $\partial^2 Q(x,a)/\partial a^2$ equals zero. In view of 2, when $\pi(x) \leq 0.5$, the bias of $\widehat{V}_h^{(1)}(\pi)$ will be small even with a sufficiently large $h$. As such, a large $h$ is preferred to reduce the variance of $\widehat{V}_h^{(1)}(\pi)$. When $\pi(x) > 0.5$, a small $h$ is preferred to reduce the bias of $\widehat{V}_h^{(2)}(\pi)$. See Table 1 for details where we report the bias and standard deviation of $\widehat{V}_h^{(1)}(\pi)$ and $\widehat{V}_h^{(2)}(\pi)$ with two different bandwidths.

Due to the use of a single bandwidth, the kernel-based estimator suffers from either a large bias or a large variance. To overcome this limitation, we propose to adaptively discretize the action space into a union of disjoint intervals such that within each interval $\mathcal{I}$, the Q-function $\{Q(x,a) : a \in \mathcal{I}\}$ can be well-approximated by some function $Q_\mathcal{I}(x)$ that is constant in $a \in \mathcal{I}$. Based on the discretization, one can apply IPW or DR to evaluate the value. The advantage of adaptive discretization is illustrated in the right panel of Figure 1. When $a \leq 0.5$, the Q-function is constant in $a$. It is likely that our procedure will not further split the interval $[0, 0.5]$. Consequently, the corresponding DR estimator for $V^{(1)}(\pi)$ will not suffer from large variance. When $a > 0.5$, our procedure will split $(0.5, 1]$ into a series of sub-intervals, approximating $Q$ by a step function. This guarantees the resulting DR estimator for $V^{(2)}(\pi)$ will not suffer from large bias. Consequently, the proposed value estimator achieves a smaller mean squared error than kernel-based estimators. See Table 1 for details.

## 3.2 THE MAIN IDEA

For simplicity, we set the action space $\mathcal{A} = [0, 1]$. From now on, we focus on a subset of intervals in $[0, 1]$. By *interval* we always refer to those of the form $[c, d)$ for some $0 \leq c < d < 1$, or $[c, 1]$ for some $0 \leq c < 1$, denoted as $\mathcal{I}$. A discretization $\mathcal{D}$ for $\mathcal{A}$ is defined as a collection of mutually disjoint intervals that covers $\mathcal{A}$. Let $|\mathcal{D}|$ denote the number of intervals in $\mathcal{D}$ and $|\mathcal{I}|$ denote the length of the interval $\mathcal{I}$. We aim to identify an "optimal" discretization $\widehat{\mathcal{D}}$ such that for each interval $\mathcal{I} \in \widehat{\mathcal{D}}$, $Q(x,a)$ is approximately a constant function of $a \in \mathcal{I}$. The number of intervals in $\widehat{\mathcal{D}}$ represents a trade-off. If $|\widehat{\mathcal{D}}|$ is too large, then $\widehat{\mathcal{D}}$ will contain many short intervals, the resulting IPW or DR estimator might suffer from large variance. Yet, a smaller value of $|\widehat{\mathcal{D}}|$ might result in a large bias. Our proposed method adaptively determines $\widehat{\mathcal{D}}$ and its size $|\widehat{\mathcal{D}}|$ as illustrated below.

To begin with, we cut the entire action space $\mathcal{A}$ into $m$ initial intervals: $[0, 1/m], [1/m, 2/m), \ldots,$ $[(m-1)/m, 1]$. The number $m$ shall be sufficiently large such that the Q-function can be well-approximated by a piecewise function on these intervals. In practice, we recommend to set the initial number of intervals $m$ to be proportional to the sample size $n$. Note the set of these initial intervals is *not* the final partition $\widehat{\mathcal{D}}$ that we recommend, but only serve as the initial candidate intervals. We next adaptively combine some of these initial intervals to form the final partition $\widehat{\mathcal{D}}$. As shown in our numerical studies (see Table 5 in Appendix B for more details), the size of the final partition $|\widehat{\mathcal{D}}|$ is usually much less than $m$.

More specifically, denote by $\mathcal{B}(m)$ as the set of discretizations $\mathcal{D}$ such that the end-points of each interval $\mathcal{I} \in \mathcal{D}$ lie on the grid $\{j/m : j = 0, 1, \cdots, m\}$. We associate to each partition $\mathcal{D} \in \mathcal{B}(m)$ a collection of functions $\{Q_\mathcal{I}\}_{\mathcal{I} \in \mathcal{D}}$. These functions depend only on features, not the action. They are used to produce a piecewise approximation of the Q-function such that $Q(a, \bullet) \approx \sum_\mathcal{I} \mathbb{I}(a \in \mathcal{I})Q_\mathcal{I}(\bullet)$. We model these $Q_\mathcal{I}$ using deep neural networks, to capture the complex dependence between the response and features. When the Q-function is well-approximated, we expect the least square loss $\sum_{\mathcal{I} \in \mathcal{D}}\left[\sum_{i=1}^{n}\mathbb{I}(A_i \in \mathcal{I})\{Y_i - Q_\mathcal{I}(X_i)\}^2\right]$, will be small.

Consequently, $\widehat{\mathcal{D}}$ can be estimated by solving

$$(\widehat{\mathcal{D}}, \{\widehat{q}_\mathcal{I} : \mathcal{I} \in \widehat{\mathcal{D}}\}) = \underset{(\mathcal{D} \in \mathcal{B}(m), \{Q_\mathcal{I} \in \mathcal{Q} : \mathcal{I} \in \mathcal{D}\})}{\arg\min} \left(\sum_{\mathcal{I} \in \mathcal{D}}\left[\frac{1}{n}\sum_{i=1}^{n}\mathbb{I}(A_i \in \mathcal{I})\{Y_i - Q_\mathcal{I}(X_i)\}^2\right] + \gamma|\mathcal{D}|\right), \quad (3)$$

for some regularization parameter $\gamma$ and DNN class $\mathcal{Q}$. Here, the penalty term $\gamma|\mathcal{D}|$ in equation 3 controls the total number of intervals in $\widehat{\mathcal{D}}$, as in multi-scale change point detection. A large $\gamma$ results in few intervals in $\widehat{\mathcal{D}}$ and a potential large bias of the value estimator, whereas a small $\gamma$

could procedure a large number of intervals in $\widehat{\mathcal{D}}$, leading to a noisy value estimator. In practice, we use cross-validation to select the regularization parameter $\gamma$ that minimizes the mean square error of the fitted Q-function. We refer to this step as deep jump Q-learning. Details of this step are given in the next section. Given $\widehat{\mathcal{D}}$, one can apply IPW or DR (see equation 1) to derive the value estimates.

To further reduce the bias of the value estimator, we employ a data splitting and cross-fitting strategy, which is commonly used in statistics (Romano & DiCiccio, 2019). That is, we use different subsets of data samples to learn the discretization $\widehat{\mathcal{D}}$ and to construct the value estimator. A pseudocode summarizing our algorithm is given in Algorithm 1 in Appendix A. We present the details below.

### 3.3 THE COMPLETE ALGORITHM

We present the details for DJQE in this section. It consists of three steps: data splitting, deep jump Q-learning, and cross-fitting.

**Step 1: Data Splitting:** We divide all $n$ samples into $\mathcal{L}$ subsets of equal size, where $\mathbb{L}_\ell$ denotes the indices of samples in the $\ell$th subset for $\ell = 1, \cdots, \mathcal{L}$. Let $\mathbb{L}_\ell^c = \{1, 2, \cdots, n\} - \mathbb{L}_\ell$ as the complement of $\mathbb{L}_\ell$.

**Step 2: Deep Jump Q-Learning:** For each $\ell = 1, \cdots, \mathcal{L}$, we propose to apply deep jump Q-learning to compute a discretization $\widehat{\mathcal{D}}^{(\ell)}$ and $\{\widehat{q}_{\mathcal{I}}^{(\ell)} : \mathcal{I} \in \widehat{\mathcal{D}}^{(\ell)}\}$ by solving a version of equation 3 using the data subset in $\mathbb{L}_\ell^c$ only. We next present the computational details for solving this optimization. Our approach is motivated by the PELT method (Killick et al., 2012) in multi-scale change point detection. Specifically, for any interval $\mathcal{I}$, define $\widehat{q}_{\mathcal{I}}^{(\ell)}$ as the minimizer of

$$\arg \min_{Q_{\mathcal{I}} \in \mathcal{Q}} \frac{1}{|\mathbb{L}_\ell^c|} \sum_{i \in \mathbb{L}_\ell^c} \mathbb{I}(A_i \in \mathcal{I}) \big\{ Q_{\mathcal{I}}(X_i) - Y_i \big\}^2, \tag{4}$$

where $|\mathbb{L}_\ell^c|$ denotes the number of samples in $\mathbb{L}_\ell^c$. Define the cost function $\mathcal{C}^{(\ell)}(\mathcal{I})$ as the minimum value of the objective function 4, i.e, $\mathcal{C}^{(\ell)}(\mathcal{I}) = \frac{1}{|\mathbb{L}_\ell^c|} \sum_{i \in \mathbb{L}_\ell^c} \mathbb{I}(A_i \in \mathcal{I}) \big\{ \widehat{q}_{\mathcal{I}}^{(\ell)}(X_i) - Y_i \big\}^2$. In our implementation, we set $\mathcal{Q}$ as the class of multilayer perceptrons (MLPs, see Figure 3 in Appendix A for an illustration) with $L$ hidden layers and $H$ nodes each hidden layer. The above optimization can be solved via the MLP regressor implementation of Pedregosa et al. (2011).

Computation of $\widehat{\mathcal{D}}^{(\ell)}$ relies on dynamic programming (Friedrich et al., 2008). For any integer $1 \leq v^* < m$, denote by $\mathcal{B}(m, v^*)$ the set consisting of all possible discretizations $\mathcal{D}_{v^*}$ of $[0, v^*/m)$. Set $\mathcal{B}(m, m) = \mathcal{B}(m)$, we define the Bellman function as

$$\text{Bell}(v^*) = \inf_{\mathcal{D}_{v^*} \in \mathcal{B}(m, v^*)} \left( \sum_{\mathcal{I} \in \mathcal{D}_{v^*}} \mathcal{C}^{(\ell)}(\mathcal{I}) + \gamma(|\mathcal{D}_{v^*}| - 1) \right), \text{ and } \text{Bell}(0) = -\gamma.$$

Our algorithm recursively updates the Bellman function based on the following formula,

$$\text{Bell}(v^*) = \min_{v \in \mathcal{R}_{v^*}} \big\{ \text{Bell}(v) + \mathcal{C}^{(\ell)}([v/m, v^*/m)) + \gamma \big\}, \quad \forall v^* \geq 1, \tag{5}$$

where $\mathcal{R}_{v^*}$ is the candidate change points list updated by

$$\{v \in \mathcal{R}_{v^*-1} \cup \{v^* - 1\} : \text{Bell}(v) + \mathcal{C}^{(\ell)}([v/m, (v^* - 1)/m)) \leq \text{Bell}(v^* - 1)\}, \tag{6}$$

during each iteration with $\mathcal{R}_0 = \{0\}$. The constraint listed in 6 is important as it facilitates the computation by discarding change points not relevant to obtain the final discretization, leading to a linear computational cost (Killick et al., 2012).

To solve equation 5, we search the optimal change point location $v$ that minimizes $\text{Bell}(v^*)$. This requires to apply the MLP regressor to learn $\widehat{q}_{[v/m, v^*/m)}^{(\ell)}$ and $\mathcal{C}^{(\ell)}([v/m, v^*/m))$ for each $v \in \mathcal{R}_{v^*}$. Let $v^1$ be the corresponding minimizer. We then define the change points list $\tau(v^*) = \{v^1, \tau(v^1)\}$. This procedure is iterated to compute $\text{Bell}(v^*)$ and $\tau(v^*)$ for $v^* = 1, \ldots, m$. The optimal partition $\widehat{\mathcal{D}}^{(\ell)}$ is determined by the values stored in $\tau$ (see Algorithm 1 in Appendix A for details).

**Step 3. Cross-Fitting:** For each interval in the estimated optimal partition $\widehat{\mathcal{D}}^{(\ell)}$, we estimate the generalized propensity score function $\Pr(A \in \mathcal{I} | X = x)$ via the MLP regressor using the training dataset $\mathbb{L}_\ell^c$. Let $\widehat{b}^{(\ell)}(\mathcal{I}|x)$ denote the resulting estimate. The final estimated value for $V(\pi)$ is

constructed via cross-fitting, given by,

$$\widehat{V}(\pi) = \frac{1}{n} \sum_{\ell=1}^{\mathcal{L}} \sum_{\mathcal{I} \in \widehat{\mathcal{D}}^{(\ell)}} \sum_{i \in \mathbb{L}_\ell} \left[ \mathbb{I}(A_i \in \mathcal{I}) \frac{\mathbb{I}\{\pi(X_i) \in \mathcal{I}\}}{\widehat{b}_{\mathcal{I}}^{(\ell)}(\mathcal{I}|X_i)} \{Y_i - \widehat{q}_{\mathcal{I}}^{(\ell)}(X_i)\} + \mathbb{I}(A_i \in \mathcal{I}) \widehat{q}_{\mathcal{I}}^{(\ell)}(X_i) \right]. \quad (7)$$

Note the samples used to construct $\widehat{V}$ inside bracket are independent from those to estimate $\widehat{q}_{\mathcal{I}}^{(\ell)}, \widehat{b}_{\mathcal{I}}^{(\ell)}$ and $\widehat{\mathcal{D}}^{(\ell)}$. This helps remove the bias induced by overfitting in the estimation of $\widehat{q}_{\mathcal{I}}^{(\ell)}, \widehat{b}_{\mathcal{I}}^{(\ell)}$ and $\widehat{\mathcal{D}}^{(\ell)}$.

## 4 THEORY

We investigate the theoretical properties of the proposed estimator. For simplicity, let the support $\mathcal{X} = [0,1]^p$. We will show our estimator is consistent when the Q-function is either a piecewise function or a continuous function of $a$. Specifically, consider the following two model assumptions.

**Model 1 (Piece-wise constant function).** Suppose

$$Q(x,a) = \sum_{\mathcal{I} \in \mathcal{D}_0} q_{\mathcal{I}}(x) \mathbb{I}(a \in \mathcal{I}), \qquad \forall x \in \mathcal{X}, \quad \forall a \in \mathcal{A}, \quad (8)$$

for some partition $\mathcal{D}_0$ of $[0,1]$ and a collection of continuous functions $(q_{\mathcal{I}})_{\mathcal{I} \in \mathcal{D}_0}$.

**Model 2 (Continuous function).** Suppose $Q$ is a continuous function of $a$ and $x$.

We first consider the case where the value function takes the form of equation 8. We remark that kernel-based estimators will fail under this model assumption, as it requires the second-order derivative of the Q-function to exist. Without loss of generality, assume $q_{\mathcal{I}_1} \neq q_{\mathcal{I}_2}$ for any two adjacent intervals $\mathcal{I}_1, \mathcal{I}_2 \in \mathcal{D}_0$. This guarantees that the representation in equation 8 is unique. For any partition $\mathcal{D} = \{[0, \tau_1), [\tau_1, \tau_2), \cdots, [\tau_K, 1]\}$, we use $J(\mathcal{D})$ to denote the set of change points $\{\tau_1, \cdots, \tau_K\}$. We impose the following conditions to establish our theories.

**Assumption 1.** The number of layers $L$ and the number of nodes in each hidden layer $H$ diverge with $n$, in that $HL = O(n^\rho)$, for some constant $0 < \rho < 1/2$.

**Assumption 2.** Functions $\{\widehat{q}_{\mathcal{I}}^{(\ell)}(\bullet)\}_{\mathcal{I} \in \widehat{\mathcal{D}}^{(\ell)}}$ are uniformly bounded.

Assumption 1 is mild, as both $H$ and $L$ are parameters we specify. The part that $HL = O(n^\rho)$ ensures that the stochastic error resulting from the parameter estimation in the MLP is negligible. Assumption 2 ensures that the optimizer would not diverge in the $\ell_\infty$ sense. Similar assumptions are commonly imposed in the literature to derive the convergence rates of DNN estimators (see e.g., Farrell et al., 2018). These two assumptions guarantee the uniform consistency of the DNN estimator $\{\widehat{q}_{\mathcal{I}}\}_{\mathcal{I} \in \widehat{\mathcal{D}}^{(\ell)}}$. See Lemma 1 in Appendix D for details.

**Theorem 1** *Suppose Model 1, Assumptions 1 and 2 hold. Suppose $m$ diverges to infinity with $n$. Then, there exists some constant $\gamma_0$ such that as long as $0 < \gamma \leq \gamma_0$, the following events occur with probability approaching 1 (w.p.a.1),*

*(i) $|\widehat{\mathcal{D}}^{(\ell)}| = |\mathcal{D}_0|$; (ii) $\max_{\tau \in J(\mathcal{D}_0)} \min_{\widehat{\tau} \in J(\widehat{\mathcal{D}}^{(\ell)})} |\widehat{\tau} - \tau| = o_p(1)$. (iii) $\widehat{V}(\pi) = V(\pi) + o_p(1)$ for any policy $\pi$ such that for any $\tau_0 \in J(\mathcal{D}_0)$, $Pr(\pi(X) \in [\tau_0 - \epsilon, \tau_0 + \epsilon]) \to 0$ as $\epsilon \to 0$.*

Theorem 1 establishes the properties of our method under settings where the $Q(a, x)$ is piecewise function in $a$. Results in (i) imply that deep jump Q-learning correctly identifies the number of change points. Results in (ii) imply that any change point in $\mathcal{D}_0$ can be consistently identified. In particular, $\widehat{\mathcal{D}}^{(\ell)}$ corresponds to a subset of $\{1/m, 2/m, \cdots, (m-1)/m\}$. With a sufficiently large $m$, for any true change point $\tau$ in $\mathcal{D}_0$, there will be a change point in $\widehat{\mathcal{D}}^{(\ell)}$ that approaches $\tau$. Consequently, the change point locations can be consistently estimated. To ensure the consistency of the proposed value estimator, we require that the distribution of the random variable $\pi(X)$ induced by the target policy does not have point-masses at the change point locations. This condition is also mild. For instance, it automatically holds when $\pi(X)$ has a density function on $[0, 1]$.

**Theorem 2** *Suppose Model 2, Assumptions 1 and 2 hold. Suppose $m$ diverges to infinity and $\gamma$ decays to zero. Then we have*

*(i) $\max_{\mathcal{I} \in \widehat{\mathcal{D}}^{(\ell)}} \sup_{a \in \mathcal{I}} \mathbb{E}|\widehat{q}_{\mathcal{I}}^{(\ell)}(X) - Q(X, a)|^2 = o(1)$; (ii) $\widehat{V}(\pi) - V(\pi) = o_p(1)$ for any $\pi$.*

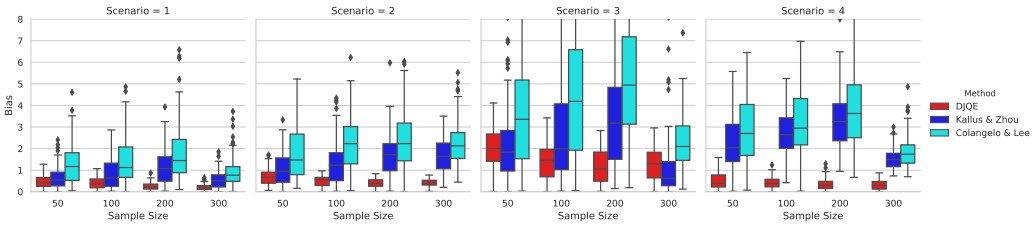

Figure 2: The box plot of the estimated values of the optimal policy under the proposed DJQE and two kernel-based methods for Scenario 1-4.

Theorem 2 establishes the properties of our method under settings where $Q$ is continuous in $a$. Results in (i) imply that $\hat{q}_{\mathcal{I}}^{(\ell)}(\bullet)$ can be used to uniformly approximate $Q(a, \bullet)$ for any $a \in \mathcal{I}$. The consistency of the value in (ii) thus follows.

Finally, we conduct in-depth theoretical analysis to demonstrate the advantage of our estimator. Due to space constraint, we present an informal statement below. Details are given in Appendix C.

**Theorem 3 (Informal Statement)**
(i) When the Q-function belongs to a class of piecewise constant functions of the action $a$, the minimax rate of convergence of the proposed value estimator is $O_p(n^{-1/2})$. In contrast, the minimax convergence rate of kernel-based estimator is $O_p(n^{-1/3})$.

(ii) When the Q-function belongs to a class of smooth functions of $a$, the minimax convergence rate of our estimator is faster than kernel-based estimators if the bandwdith undersmoothes or oversmoothes the data.

## 5 EXPERIMENTS

In this section, we investigate the finite sample performance of the proposed DJQE on the synthetic and real datasets, in comparison to two kernel-based methods. The computing infrastructure used is a virtual machine in the AWS Platform with 72 processor cores and 144GB memory.

### 5.1 SYNTHETIC DATA

Synthetic data are generated from the following model:

$$Y|X, A \sim N\{Q(X, A), 1\}, \quad b(A|X) \sim \text{Unif}[0, 1] \quad \text{and} \quad X^{(1)}, X^{(2)}, \ldots, X^{(p)} \overset{iid}{\sim} \text{Unif}[-1, 1],$$

where $X = [X^{(1)}, X^{(2)}, \ldots, X^{(p)}]$. Consider the following different scenarios:
**S1**: $Q(x, a) = (1 + x^{(1)})\mathbb{I}(a < 0.35) + (x^{(1)} - x^{(2)})\mathbb{I}(0.35 \leq a < 0.65) + (1 - x^{(2)})\mathbb{I}(a \geq 0.65)$;
**S2**: $Q(x, a) = \mathbb{I}(a < 0.25) + \sin(2\pi x^{(1)})\mathbb{I}(0.25 \leq a < 0.5)\{0.5 - 8(x^{(1)} - 0.75)^2\}\mathbb{I}(0.5 \leq a < 0.75) + 0.5\mathbb{I}(a \geq 0.75)$;
**S3 (toy)**: $Q(x, a) = 10 \max\{a^2 - 0.25, 0\} \log(x^{(1)} + 2)$;
**S4**: $Q(x, a) = 0.2(8 + 4x^{(1)} - 2x^{(2)} - 2x^{(3)}) - 2(1 + 0.5x^{(1)} + 0.5x^{(2)} - 2a)^2$.

The Q-function is a piecewise function of $a$ under Scenarios 1 and 2, and is continuous under Scenarios 3 (toy example considered in Section 3.1) and 4. We set the target policy to be the optimal policy that achieves the highest possible mean reward. We list the oracle mean value under the optimal policy for each scenario in the first column of Table 4 in Appendix B.

We apply the proposed DJQE and two kernel-based methods (Kallus & Zhou, 2018; Colangelo & Lee, 2020) to Scenario 1-4 with 20-dimensional covariates ($p = 20$) and $n \in \{50, 100, 200, 300\}$. For the DJQE, we select $\gamma \in \{0.1, 0.2, 0.3, 0.4, 0.5\}n^{0.4}$ based on five-fold cross-validation. Here, we set $m = n/10$ to achieve a good balance between the bias and the computational cost (see Figure 4 in Appendix B for the detailed computational cost of the DJQE and the resulting bias as a function of $m$ in Scenario 1 with $n = 100$). We find it extremely computationally intensive to compute the optimal bandwidth $h^*$ using Kallus & Zhou (2018)'s method (see the detailed comparison of computational cost under different methods based on Scenario 1 in Table 3). Thus, as suggested in Kallus & Zhou (2018), we first compute $h^*$ using data with sample size $n_0 = 50$. To accommodate data with different sample sizes $n$, we adjust $h^*$ by setting $h^*\{n_0/n\}^{0.2}$. To implement Colangelo

Table 2: The bias, the standard deviation, and the mean squared error of the estimated values under the optimal policy via the proposed DJQE and two kernel-based methods for the Warfarin data.

| Methods | Bias | Standard deviation | Mean squared error |
|---------|------|--------------------|--------------------|
| DJQE | 0.259 | 0.416 | 0.240 |
| Kallus & Zhou (2018) | 0.662 | 0.742 | 0.989 |
| Colangelo & Lee (2020) | 0.442 | 1.164 | 1.550 |

& Lee (2020)'s estimator, we consider a list of bandwidths, given by $h = c\sigma_A n^{-0.2}$ with $c \in \{0.5, 0.75, 1.0, 1.5\}$ and $\sigma_A$ is the sample standard deviation of the action. We then manually select the best bandwidth such that the resulting value estimator achieves the smallest mean squared error. The conditional mean value and generalized propensity score are fitted via the MLP regressor with 10 hidden layer and 10 neurons in each layer. The average estimated value and its standard deviation over 100 replicates are illustrated in Figure 2 for different methods, with detailed values reported in Table 4 in Appendix B. In addition, we provide the size of the final estimated partition under the DJQE in Table 5 in Appendix B, which is much smaller than $m$ in most cases.

It can be seen from Figure 2 that the proposed DJQE is very efficient and outperforms all competing methods in almost all cases. We note that the proposed method performs reasonably well even when the sample size is small ($n = 50$). In contrast, two kernel-based methods fail to accurately estimate the value even in some cases when $n = 300$. Among the two kernel-based OPE approaches, we observe that the method developed by Kallus & Zhou (2018) performs better in general.

## 5.2 REAL DATA: PERSONALIZED DOSE FINDING

Warfarin is commonly used for preventing thrombosis and thromboembolism. We use the dataset provided by the International Warfarin Pharmacogenetics (Consortium, 2009) for analysis. We choose $p = 81$ baseline covariates considered in Kallus & Zhou (2018). This yields a total of 3964 with complete records of baseline information. The response is defined as the absolute distance between the international normalized ratio (INR, a measurement of the time it takes for the blood to clot) after the treatment and the ideal value 2.5, i.e, $Y = -|\text{INR} - 2.5|$. We use the min-max normalization to convert the range of the dose level $A$ into $[0, 1]$.

To compare among different methods, we calibrate the dataset to generate simulated outcomes. Specifically, we first estimate the Q-function via a MLP regressor with 10 hidden layers and 50 neurons in each layer using the whole dataset. The goodness-of-the-fit of the fitted model under the MLP regressor is reported in Table 6 in Appendix B. We next use the fitted Q-function $\widehat{Q}(X, A)$ to simulate the data. Given a randomly sampled feature-action pair $(a_j, x_j)$ from $\{(A_1, X_1), \cdots, (A_n, X_n)\}$, we set the reward $r_j$ to $N\{\widehat{Q}(x_j, a_j), \widehat{\sigma}^2\}$, where $\widehat{\sigma}$ is the standard deviation of the fitted residual $\{Y_i - \widehat{Q}(X_i, A_i)\}_i$. Given the simulated data $\{(x_j, a_j, r_j) : 1 \leq j \leq n\}$, we are interested in evaluating the optimal policy: $\pi^\star(X) \equiv \arg\max_{a \in [0,1]} \widehat{Q}(X, a)$. The oracle value under the optimal policy is $V = -0.278$.

We apply the DJQE on the calibrated Warfarin data, against two kernel-based methods. Due to the extremely intensive computation in Kallus & Zhou (2018), we directly apply the estimated optimal bandwidth $h^*$ in their real data analysis, since they used the same dataset. Biases, standard deviations, and mean squared errors of the estimated values are reported in Table 2 over 20 replicates for sample size $n = 500$ under different methods.

It can be observed from Table 2 that our proposed DJQE achieves much smaller mean squared error than the two kernel-based methods, when evaluating the optimal policy. Specifically, the DJQE yields bias as 0.259 with the standard deviation as 0.416, in contrast to the large bias as 0.662 with the standard deviation as 0.742 under Kallus & Zhou (2018)'s method and the bias as 0.442 with the large standard deviation as 1.164 under Colangelo & Lee (2020)'s method. Therefore, our proposed DJQE for off-policy evaluation with continuous actions works better than the kernel-based methods.

## 6 DISCUSSION

Currently, we focus on settings with a single decision point. It would be practically interesting to extend our proposal to sequential decision making. A potential drawback of our method is that it would be very computationally intensive for a large $m$ as the runtime increases linearly in $m$.

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

## A   MORE ON THE IMPLEMENTATION

We summarize our algorithm in Algorithm 1.

**Global:** data $\{(X_i, A_i, Y_i)\}_{1 \leq i \leq n}$; number of initial intervals $m$; penalty term $\gamma$; target policy $\pi$.
**Local:** an upper triangular matrix of cost $\mathcal{C} \in \mathbb{R}^{m(m+1)/2}$; Bellman function Bell $\in \mathbb{R}^m$; partitions $\widehat{\mathcal{D}}$;
  DNN functions $\{\widehat{q}_{\mathcal{I}}, \widehat{b}_{\mathcal{I}} : \mathcal{I} \in \widehat{\mathcal{D}}\}$; a vector $\tau \in \mathbb{N}^m$; a set of candidate point lists $\mathcal{R}$.
**Output:** the value estimator for target policy $\widehat{V}(\pi)$.
I. Split all $n$ samples into $L$ subsets as $\{\mathbb{L}_1, \cdots, \mathbb{L}_L\}$; $\widehat{V}(\pi) \leftarrow 0$;
II. Initialize an even segment on the action space with $m$ pieces:
  $\{\mathcal{I}\} = \{[0, 1/m), [1/m, 2/m), \ldots, [(m-1)/m, 1]\}$;
III. For $\ell = 1, \cdots, L$:
 1. Set the training dataset as $\mathbb{L}_\ell^c = \{1, 2, \cdots, n\} - \mathbb{L}_\ell$;
 2. Bell$(0) \leftarrow -\gamma$; $\widehat{\mathcal{D}} = [0, 1]$; $\tau \leftarrow Null$; $\mathcal{R}(0) \leftarrow \{0\}$;
 3. Collect cost function:
 For $r = 1, \ldots, m$: For $l = 0, \ldots, (r-1)$:
  (i). Let $\mathcal{I} = [l/m, r/m)$ if $r < m$ else $\mathcal{I} = [l/m, 1]$;
  (ii). Fit a MLP regressor: $\widehat{q}_{\mathcal{I}}(\cdot) \leftarrow \mathbb{I}(i \in \mathbb{L}_\ell^c)\mathbb{I}(A_i \in \mathcal{I})Y_i \sim \mathbb{I}(A_i \in \mathcal{I})MLP(X_i)$;
  (iii). Calculate the cost: $\mathcal{C}(\mathcal{I}) \leftarrow \sum_{i \in \mathbb{L}_\ell^c} \mathbb{I}(A_i \in \mathcal{I})\{\widehat{q}_{\mathcal{I}}(X_i) - Y_i\}^2$;
 4. Apply the PELT method to get partitions: For $v^* = 1, \ldots, m$:
  (i).Bell$(v^*) = \min_{v \in \mathcal{R}(v^*)}\{$Bell$(v) + \mathcal{C}([v/m, v^*/m)) + \gamma\}$;
  (ii). $v^1 \leftarrow \arg\min_{v \in \mathcal{R}(v^*)}\{$Bell$(v) + \mathcal{C}([v/m, v^*/m)) + \gamma\}$;
  (iii). $\tau(v^*) \leftarrow \{v^1, \tau(v^1)\}$;
  (iv). $\mathcal{R}(v^*) \leftarrow \{v \in \mathcal{R}(v^* - 1) \cup \{v^* - 1\} :$ Bell$(v) + \mathcal{C}([v/m, (v^* - 1)/m)) \leq$ Bell$(v^* - 1)\}$;
 5. Construct the DR value estimator: $r \leftarrow m$; $l \leftarrow \tau[r]$; While $r > 0$:
  (i) Let $\mathcal{I} = [l/m, r/m)$ if $r < m$ else $\mathcal{I} = [l/m, 1]$; $\widehat{\mathcal{D}} \leftarrow \widehat{\mathcal{D}} \cup \mathcal{I}$;
  (ii) Recall fitted MLP: $\widehat{q}_{\mathcal{I}}(\cdot) \leftarrow \mathbb{I}(i \in \mathbb{L}_\ell^c)\mathbb{I}(A_i \in \mathcal{I})Y_i \sim \mathbb{I}(A_i \in \mathcal{I})MLP(X_i)$;
  (iii) Fit propensity score: $\widehat{b}_{\mathcal{I}}(\cdot) \leftarrow \mathbb{I}(i \in \mathbb{L}_\ell^c)\mathbb{I}(A_i \in \mathcal{I}) \sim \mathbb{I}(A_i \in \mathcal{I})MLP(X_i)$;
  (iv) $r \leftarrow l$; $l \leftarrow \tau(r)$;
 6. Evaluation using testing dataset $\mathbb{L}_\ell$:
  $\widehat{V}(\pi) += \sum_{\mathcal{I} \in \widehat{\mathcal{D}}} \left( \sum_{i \in \mathbb{L}_\ell} \mathbb{I}(A_i \in \mathcal{I}) \left[ \frac{\mathbb{I}\{\pi(X_i) \in \mathcal{I}\}}{\widehat{b}_{\mathcal{I}}(\mathcal{I}|X_i)}\{Y_i - \widehat{q}_{\mathcal{I}}(X_i)\} + \widehat{q}_{\mathcal{I}}(X_i) \right] \right)$;
**return** $\widehat{V}(\pi)/n$ .

**Algorithm 1:** Deep Jump Q-Evaluation

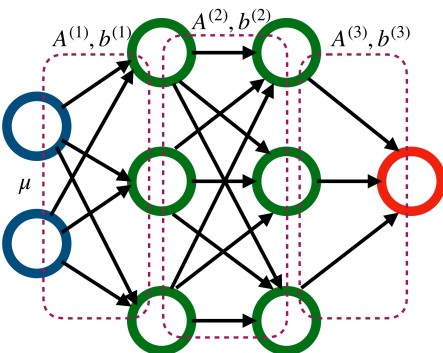

Figure 3: Illustration of multilayer perceptron with two hidden layers and three nodes each hidden layer. Here $\mu$ is the input, $A^{(l)}$ and $b^{(l)}$ denote the corresponding parameters to produce the linear transformation for the $(l-1)$th layer.

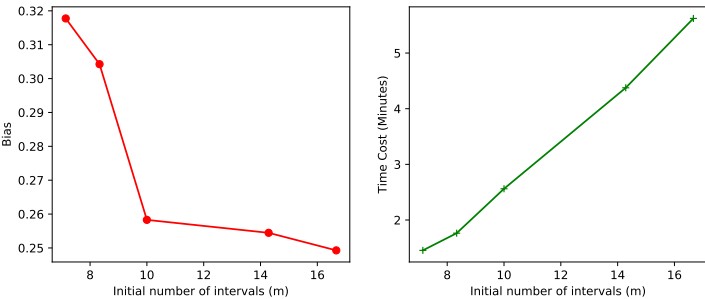

Figure 4: The bias of the estimated value and the computational cost (in minutes) under the DJQE with different initial number of intervals ($m$) when $n = 100$ in Scenario 1.

## B    ADDITIONAL EXPERIMENTAL RESULTS

We include additional experimental results in this section.

Table 3: The averaged computational cost (in minutes) under the proposed DJQE and two kernel-based methods for Scenario 1.

| Method | DJQE | Kallus & Zhou (2018) | Colangelo & Lee (2020) |
|---|---|---|---|
| $n = 50$ | $< 1$ | 365 | $< 1$ |
| $n = 100$ | 3 | 773 | $< 1$ |
| $n = 200$ | 7 | $> 1440$ (24 hours) | $< 1$ |
| $n = 300$ | 14 | $> 2880$ (48 hours) | $< 1$ |

Table 4: The bias and the standard deviation (in parentheses) of the estimated values of the optimal policy under the proposed DJQE and two kernel-based methods for Scenario 1 to 4.

|  | $n$ | 50 | 100 | 200 | 300 |
|---|---|---|---|---|---|
| Scenario 1 | DJQE | 0.445(0.381) | 0.398(0.391) | 0.253(0.269) | 0.209(0.210) |
| $V = 1.33$ | Kallus & Zhou (2018) | 0.656(0.787) | 0.848(0.799) | 1.163(0.884) | 0.537(0.422) |
|  | Colangelo & Lee (2020) | 1.285(1.230) | 1.473(1.304) | 1.826(1.463) | 0.934(0.730) |
| Scenario 2 | DJQE | 0.696(0.376) | 0.502(0.311) | 0.400(0.219) | 0.411(0.168) |
| $V = 1.00$ | Kallus & Zhou (2018) | 1.061(1.124) | 1.363(1.131) | 1.679(1.032) | 1.664(0.792) |
|  | Colangelo & Lee (2020) | 1.827(1.371) | 2.292(1.458) | 2.429(1.541) | 2.264(1.062) |
| Scenario 3 | DJQE | 2.014(0.865) | 1.410(0.987) | 1.184(0.967) | 1.267(0.933) |
| $V = 4.86$ | Kallus & Zhou (2018) | 2.196(2.369) | 2.758(2.510) | 3.573(2.862) | 1.151(1.798) |
|  | Colangelo & Lee (2020) | 2.586(2.825) | 3.172(3.027) | 3.949(3.391) | 1.367(2.110) |
| Scenario 4 | DJQE | 0.494(0.485) | 0.412(0.426) | 0.349(0.383) | 0.321(0.315) |
| $V = 1.60$ | Kallus & Zhou (2018) | 2.192(1.210) | 2.740(1.034) | 3.354(1.324) | 1.555(0.500) |
|  | Colangelo & Lee (2020) | 2.975(1.789) | 3.282(1.525) | 3.921(1.927) | 1.853(0.751) |

Table 5: The averaged size of the final estimated partition ($|\widehat{\mathcal{D}}|$) in comparison to the initial number of intervals ($m$) under the proposed DJQE for Scenario 1 to 4.

| $\|\widehat{\mathcal{D}}\| / m$ | $n = 50$ | $n = 100$ | $n = 200$ | $n = 300$ |
|---|---|---|---|---|
| Scenario 1 | 3 / 5 | 4 / 10 | 6 / 20 | 6 / 30 |
| Scenario 2 | 4 / 5 | 6 / 10 | 9 / 20 | 11 / 30 |
| Scenario 3 | 4 / 5 | 6 / 10 | 8 / 20 | 10 / 30 |
| Scenario 4 | 4 / 5 | 6 / 10 | 8 / 20 | 10 / 30 |

Table 6: The mean squared error (MSE)[5], the normalized root-mean-square-deviation (NRMSD)[6], the mean absolute error (MAE)[7], and the normalized MAE (NMAE)[8]of the fitted model under the MLP regressor, linear regression, and the random forest algorithm, via ten-fold cross-validation.

| Method | MLP Regressor | Linear Regression | Random Forest |
|--------|---------------|-------------------|---------------|
| MSE    | 0.06          | 0.09              | 0.08          |
| NRMSD  | 0.13          | 0.16              | 0.15          |
| MAE    | 0.19          | 0.23              | 0.22          |
| NMAE   | 0.10          | 0.12              | 0.12          |

## C    ADDITIONAL THEORETICAL RESULTS

In this section, we conduct in-depth theoretical analysis to compare the minimax convergence rate of the proposed value estimator with kernel-based value estimator.

We first briefly summarize our theoretical findings. When the Q-function belongs to a class of piecewise constant functions of the action $a$, the proposed estimator converges at a faster rate than kernel-based estimators. Specifically, the kernel-based OPE converges at a rate of $n^{-1/3}$ whereas our estimator converges at a rate of $n^{-1/2}$.

When the Q-function belongs to a class of Lipschitz continuous functions of $a$, our estimator converges at a rate of $n^{-1/5}$ whereas kernel-based estimators converge at a slower rate when the bandwidth undersmoothes or oversmoothes the data.

As we have commented, it remains extremely challenging to tune the bandwidth of kernel-based OPE in practice, since OPE is an unsupervised problem. The use of a suboptimal bandwidth would lead to a slower rate of convergence for kernel-based estimators. In contrast, we develop a supervised learning algorithm to adaptively discretize the action space. The tuning parameters in our procedure can be selected via cross-validation.

All the above findings are supported by our observations in the toy example (Section 3.1) and numerical experiments (Section 5) as well.

We next present our theoretical results. We first introduce some notations. Define the following classes of Q-functions:

$$\mathcal{Q}_1 = \left\{ Q : Q(a,x) = \sum_{\mathcal{I} \in \mathcal{D}_0} \sum_{a \in \mathcal{I}} \mathbb{I}(a \in \mathcal{I}) q_{\mathcal{I}}(x), \max_{\substack{\mathcal{I}_1 \neq \mathcal{I}_2 \\ \mathcal{I}_1, \mathcal{I}_2 \in \mathcal{D}_0}} \mathbb{E}|q_{\mathcal{I}_1}(X) - q_{\mathcal{I}_2}(X)|^2 \geq \epsilon, \right.$$
$$\left. |\mathcal{D}_0| \leq C_1, Q(a,\cdot) \in \Lambda(\beta, C_2), \forall a \right\}, \tag{9}$$
$$\mathcal{Q}_2 = \left\{ Q : \sup_{a_1,a_2,x} |Q(a_1,x) - Q(a_2,x)| \leq C_3, Q(a,\cdot) \in \Lambda(\beta, C_2), \forall a \right\},$$

for some sufficiently small constant $\epsilon > 0$ and some sufficiently large constants $C_1, C_2 > 0$ where $\Lambda(\beta, C_2)$ denotes the class of $\beta$-smooth functions (see e.g., Section 3.1.1, Shi et al., 2020). By definition, the first set $\mathcal{Q}_1$ consists of all piecewise constant functions of $a$ with at most $C_1 - 1$ change points. The second set $\mathcal{Q}_2$ contains the class of Lipschitz continuous functions of $a$.

Finally, define the policy class $\Pi = \{\pi : \pi(X) \text{ has a density function bounded by } C_4\}$ for some sufficiently large constant $C_4 > 0$. To simplify the analysis, we assume the behavior policy is

---

[5]$MSE = \frac{1}{n} \sum_{i=1}^{n} (Y_i - \widehat{Y}_i)^2$. See https://en.wikipedia.org/wiki/Mean_squared_error.

[6]$NRMSD = \frac{\sqrt{MSE}}{\max(Y) - \min(Y)}$. See https://en.wikipedia.org/wiki/Root-mean-square_deviation.

[7]$MAE = \frac{1}{n} \sum_{i=1}^{n} |Y_i - \widehat{Y}_i|$. See https://en.wikipedia.org/wiki/Mean_absolute_error.

[8]$NMAE = \frac{MAE}{\max(Y) - \min(Y)}$. See https://en.wikipedia.org/wiki/Root-mean-square_deviation.

known such that $\widehat{b} = \widehat{b}_{\mathcal{I}}^{(\ell)} = b$. In addition, assume the conditional variance of $Y$ given $A$ and $X$ is uniformly bounded away from zero.

**Theorem 3** *Suppose $2\beta > p$. Then for any $\pi \in \Pi$ and any $Q \in \mathcal{Q}_1$, with proper choice of $\gamma$ and the MLP class, the minimax rate of convergence of the proposed value estimator is $O_p(n^{-1/2})$. In contrast, the minimax convergence rate of kernel-based estimator is $O_p(n^{-1/3})$.*

*Suppose $4\beta > 3p$. For any $Q \in \mathcal{Q}_2$, with proper choice of $\gamma$ and the MLP class, the minimax rate of convergence of the proposed value estimator is $O_p(n^{-1/5})$, up to some logarithmic factors. In contrast, the minimax convergence rate of kernel-based estimator is slower when $h \asymp n^{-\kappa}$ for some $\kappa < 1/5$ or $\kappa > 3/5$.*

We briefly comment on the condition $2\beta > p$. This condition allows that the MLP regressor to converge at a rate faster than $n^{-1/4}$ (see e.g., Imaizumi & Fukumizu, 2019). Such a condition is commonly assumed in the literature on evaluating average treatment effects (see e.g., Chernozhukov et al., 2017; Farrell et al., 2018). In the second part of Theorem 3, we further require the MLP regressor to converge at a rate faster than $n^{-3/10}$.

## D   TECHNICAL PROOF

Throughout the proof, we use $c, C, c_0, \bar{c}$, etc., to denote some universal constants whose values are allowed to change from place to place. Let $O_i = \{X_i, Y_i\}$ denote the data summarized from the $i$th observation. For any interval $\mathcal{I}$, let $b(\mathcal{I}|x)$ denote the integral $\int_{a \in \mathcal{I}} b(a|x)da$.

Proofs of Theorems 1 and 2 rely on Lemmas 1 and 2. Specifically, Lemma 1 establishes the uniform convergence rate of $\widehat{q}_{\mathcal{I}}$ for any $\mathcal{I}$ whose length is no shorter than $o(\gamma)$ and belongs to the set of intervals:

$$\begin{aligned} \mathfrak{I}(m) \;=\; & \{[i_1/m, i_2/m) : \text{for some integers } i_1 \text{ and } i_2 \text{ that satisfy } 0 \le i_1 < i_2 < m\} \\ \cup \; & \{[i_3/m, 1] : \text{for some integers } i_3 \text{ that satisfy } 0 \le i_3 < m\}. \end{aligned}$$

To state this lemma, we first introduce some notations. For any such interval $\mathcal{I}$, define the function $q_{\mathcal{I}}(x) = \mathbb{E}(Y|A \in \mathcal{I}, X = x)$. It is immediate to see that the definition of $q_{\mathcal{I}}$ here is consistent with the one defined in equation 8 for any $\mathcal{I} \subseteq \mathcal{D}_0$.

**Lemma 1** *The following holds when either the conditions in Theorem 1 or 2 hold:*

$$\max_{\mathcal{I} \in \mathfrak{I}(m), |\mathcal{I}| \ge c\gamma} \mathbb{E}[|q_{\mathcal{I}}(X) - \widehat{q}_{\mathcal{I}}^{(\ell)}(X)|^2 | \{O_i\}_{i \in \mathbb{L}_\ell}] = o_p(1),$$

*for any positive constant $c > 0$. x*

**Lemma 2** *When either the conditions in Theorem 1 or 2 hold, $\min_{\mathcal{I} \in \widehat{\mathcal{D}}^{(\ell)}} |\mathcal{I}| \ge C\gamma$ w.p.a.1 for some constant $C > 0$.*

We first present the proofs for these two lemmas. Next we present the proofs for Theorems 1 and 2.

### D.1   PROOF OF LEMMA 1

For any sufficiently small constant $\epsilon > 0$, it suffices to show

$$\max_{\mathcal{I} \in \mathfrak{I}(m), |\mathcal{I}| \ge c\gamma} \mathbb{E}[|q_{\mathcal{I}}(X) - \widehat{q}_{\mathcal{I}}^{(\ell)}(X)|^2 | \{O_i\}_{i \in \mathbb{L}_\ell}] \le C\epsilon^2,$$

w.p.a.1, for some constant $C > 1$ whose value will be specified later. The proof is divided into two parts. In the first part, we show there exist some constants $\{\theta_{\mathcal{I}}^*\}_{\mathcal{I}}$ such that

$$\max_{\mathcal{I} \in \mathfrak{I}(m)} \max_{x \in \mathcal{X}} |q_{\mathcal{I}}(x) - \mathrm{MLP}(x; \theta_{\mathcal{I}}^*)| \le \epsilon. \tag{10}$$

In the second part, we show

$$\max_{\mathcal{I} \in \mathfrak{I}(m), |\mathcal{I}| \ge c\gamma} \mathbb{E}[|\mathrm{MLP}(X; \theta_{\mathcal{I}}^*) - \widehat{q}_{\mathcal{I}}^{(\ell)}(X)|^2 | \{O_i\}_{i \in \mathbb{L}_\ell}] \le (C-1)\epsilon^2. \tag{11}$$

The proof is hence completed.

**Part 1.** Under the step model assumption, for any $\mathcal{I}_0 \in \mathcal{D}_0$, $q_{\mathcal{I}_0}(\bullet)$ is continuous. Since the support $\mathcal{X}$ is compact, $q_{\mathcal{I}_0}(\bullet)$ is uniformly continuous. Similarly, we can show $b$ is uniformly continuous as well. For any $\mathcal{I} \in \mathfrak{I}_m$, we have

$$q_{\mathcal{I}}(x) = \frac{\int_{a \in \mathcal{I}} \sum_{\mathcal{I}_0 \in \mathcal{D}_0} b(a|x) \mathbb{I}(a \in \mathcal{I}_0) q_{\mathcal{I}_0}(x) da}{b(\mathcal{I}|x)}.$$

It follows from the uniform continuity of $q_{\mathcal{I}_0}$, $b$ and the positivity assumption that $q_{\mathcal{I}}(\bullet)$ is continuous for any $\mathcal{I} \in \mathfrak{I}(m)$.

Similarly, under the model assumption in Theorem 2, we can show $q_{\mathcal{I}}(\bullet)$ is continuous for any $\mathcal{I} \in \mathfrak{I}(m)$ as well. Consequently, when either the conditions in Theorem 1 or 2 hold, we obtain that $q_{\mathcal{I}}(\bullet)$ is continuous for any $\mathcal{I} \in \mathfrak{I}(m)$.

By Stone-Weierstrass theorem, there exists a multivariate polynomial function $q_{\mathcal{I}}^*$ such that the absolute value of the residual $q_{\mathcal{I}} - q_{\mathcal{I}}^*$ is uniformly bounded by $\epsilon/2$. By Theorem 1 of Yarotsky (2017), there exists a feedforward neural network with a bounded number of hidden units that uniformly approximates $q_{\mathcal{I}}^*$, with the approximation error uniformly bounded by $\epsilon/2$ in absolute value. By Lemma 1 of Farrell et al. (2018), such a feedforward network can be embedded into an MLP with a bounded number of hidden units. Since we allow $H$ and $L$ to diverge, such an MLP can be further embedded into an MLP with $L$ layers and the widths of all layers being proportional to $H$. This yields equation 10. The proof for Part 1 is thus completed.

**Part 2.** We aim to show equation 11 holds. Under the boundedness assumption on $Y$, $\{q_{\mathcal{I}}(\bullet)\}_{\mathcal{I} \in \mathfrak{I}(m)}$ are uniformly bounded. We first observe that $\mathrm{MLP}(\bullet, \theta_{\mathcal{I}}^*)$ is a bounded function, by equation 10 and the fact that $q_{\mathcal{I}}(\bullet)$ is bounded.

Next, it follows from equation 10 that

$$\frac{\mathbb{E}\mathbb{I}(A \in \mathcal{I})\{Y - \mathrm{MLP}(X; \theta_{\mathcal{I}}^*)\}^2}{\mathrm{Pr}(A \in \mathcal{I})} = \frac{\mathbb{E}\mathbb{I}(A \in \mathcal{I})\{Y - \mathbb{E}(Y|A \in \mathcal{I}, X)\}^2}{\mathrm{Pr}(A \in \mathcal{I})}$$
$$+ \frac{\mathbb{E}\mathbb{I}(A \in \mathcal{I})\{\mathbb{E}(Y|A \in \mathcal{I}, X) - \mathrm{MLP}(X; \theta_{\mathcal{I}}^*)\}^2}{\mathrm{Pr}(A \in \mathcal{I})} \leq \frac{\mathbb{E}\mathbb{I}(A \in \mathcal{I})\{Y - \mathbb{E}(Y|A \in \mathcal{I}, X)\}^2}{\mathrm{Pr}(A \in \mathcal{I})} + \epsilon^2. \tag{12}$$

By definition,

$$\frac{\sum_{i \in \mathbb{L}_\ell} \mathbb{I}(A_i \in \mathcal{I})\{Y_i - \mathrm{MLP}(X_i; \theta_{\mathcal{I}}^*)\}^2}{\mathrm{Pr}(A \in \mathcal{I})} \geq \frac{\sum_{i \in \mathbb{L}_\ell} \mathbb{I}(A_i \in \mathcal{I})\{Y_i - \mathrm{MLP}(X_i; \widehat{\theta}_{\mathcal{I}}^{(\ell)})\}^2}{\mathrm{Pr}(A \in \mathcal{I})}. \tag{13}$$

Suppose we can show

$$\sup_{\substack{\mathcal{I} \in \mathfrak{I}(m), |\mathcal{I}| \geq c\gamma \\ \theta_{\mathcal{I}} \in \Theta^*}} \left| \frac{\sum_{i \in \mathbb{L}_\ell} \mathbb{I}(A_i \in \mathcal{I})\{Y_i - \mathrm{MLP}(X_i; \theta_{\mathcal{I}})\}^2}{|\mathbb{L}_\ell| \mathrm{Pr}(A \in \mathcal{I})} - \frac{\mathbb{E}\mathbb{I}(A \in \mathcal{I})\{Y - \mathrm{MLP}(X; \theta_{\mathcal{I}})\}^2}{\mathrm{Pr}(A \in \mathcal{I})} \right| \tag{14}$$
$$= o_p(1),$$

where the set $\Theta^*$ consists of all $\theta_{\mathcal{I}}$ such that $\sup_x |\mathrm{MLP}(x; \theta_{\mathcal{I}})| \leq M$ for some sufficiently large constant $M$ such that both $\theta_{\mathcal{I}}^*$ and $\widehat{\theta}_{\mathcal{I}}^{(\ell)}$ satisfy this constraint. It thus follows from equation 12 and equation 13 that

$$\frac{\mathbb{E}\mathbb{I}(A \in \mathcal{I})\{Y - \mathrm{MLP}(X; \widehat{\theta}_{\mathcal{I}}^{(\ell)})\}^2}{\mathrm{Pr}(A \in \mathcal{I})} \leq \frac{\mathbb{E}\mathbb{I}(A \in \mathcal{I})\{Y - \mathbb{E}(Y|A \in \mathcal{I}, X)\}^2}{\mathrm{Pr}(A \in \mathcal{I})} + 2\epsilon^2,$$

w.p.a.1. Similar to equation 12, we have

$$\frac{\mathbb{E}\mathbb{I}(A \in \mathcal{I})\{Y - \mathrm{MLP}(X; \widehat{\theta}_{\mathcal{I}}^{(\ell)})\}^2}{\mathrm{Pr}(A \in \mathcal{I})} = \frac{\mathbb{E}\mathbb{I}(A \in \mathcal{I})\{Y - \mathbb{E}(Y|A \in \mathcal{I}, X)\}^2}{\mathrm{Pr}(A \in \mathcal{I})}$$
$$+ \frac{\mathbb{E}\mathbb{I}(A \in \mathcal{I})\{\mathbb{E}(Y|A \in \mathcal{I}, X) - \mathrm{MLP}(X; \widehat{\theta}_{\mathcal{I}}^{(\ell)})\}^2}{\mathrm{Pr}(A \in \mathcal{I})}.$$

It follows that

$$\frac{\mathbb{E}\mathbb{I}(A \in \mathcal{I})\{\mathbb{E}(Y|A \in \mathcal{I}, X) - \mathrm{MLP}(X; \widehat{\theta}_{\mathcal{I}}^{(\ell)})\}^2}{\mathrm{Pr}(A \in \mathcal{I})} = \frac{\mathbb{E}b(\mathcal{I}|X)\{\mathbb{E}(Y|A \in \mathcal{I}, X) - \mathrm{MLP}(X; \widehat{\theta}_{\mathcal{I}}^{(\ell)})\}^2}{\mathrm{Pr}(A \in \mathcal{I})}$$

$$\leq 2\epsilon^2.$$

Under the positivity assumption, $\sup_x b(\mathcal{I}|x)/\mathrm{Pr}(A \in \mathcal{I}) \geq 2(c-1)^{-1}$ for some constant $c > 1$. This yields equation 11.

To complete the proof, it remains to show equation 14 holds.

Using similar arguments in Section A.2.2 of Farrell et al. (2018), for any $\mathcal{I}$, we have with probability at least $1 - 2\exp(-\bar{\gamma})$ that

$$\left| \frac{\sum_{i \in \mathbb{L}_\ell} \mathbb{I}(A_i \in \mathcal{I})\{Y_i - \mathrm{MLP}(X_i; \theta_{\mathcal{I}})\}^2}{|\mathbb{L}_\ell|\mathrm{Pr}(A \in \mathcal{I})} - \frac{\mathbb{E}\mathbb{I}(A \in \mathcal{I})\{Y - \mathrm{MLP}(X; \theta_{\mathcal{I}})\}^2}{\mathrm{Pr}(A \in \mathcal{I})} \right|$$

$$\leq \bar{c}\left\{ \frac{M\bar{\gamma}}{n} + \sqrt{\frac{\bar{\gamma}}{n}}\sigma(\mathcal{I}, \theta_{\mathcal{I}}) + \sigma(\mathcal{I}, \theta_{\mathcal{I}})\sqrt{\frac{H^2 L^2}{n}}\left( \log \frac{M}{\sigma(\mathcal{I}, \theta_{\mathcal{I}})} + \log n \right) \right\},$$

for some constant $\bar{c} > 0$, where $\sigma^2(\mathcal{I}, \theta)$ corresponds to the variance of

$$\frac{1}{\mathrm{Pr}(A \in \mathcal{I})}\mathbb{I}(A \in \mathcal{I})\{Y - \mathrm{MLP}(X; \theta)\}^2.$$

Under the positivity assumption, we have $\sigma^2(\mathcal{I}, \theta) \leq O(1)|\mathcal{I}|^{-1}$ where $O(1)$ denotes some positive constant. Consequently, for any $\mathcal{I}$ whose length is greater than $c\gamma^{-1}$, we have with probability at least $1 - 2\exp(-\bar{\gamma})$ that

$$\left| \frac{\sum_{i \in \mathbb{L}_\ell} \mathbb{I}(A_i \in \mathcal{I})\{Y_i - \mathrm{MLP}(X_i; \theta_{\mathcal{I}})\}^2}{|\mathbb{L}_\ell|\mathrm{Pr}(A \in \mathcal{I})} - \frac{\mathbb{E}\mathbb{I}(A \in \mathcal{I})\{Y - \mathrm{MLP}(X; \theta_{\mathcal{I}})\}^2}{\mathrm{Pr}(A \in \mathcal{I})} \right|$$

$$\leq \bar{c}_0\left\{ \frac{M\bar{\gamma}}{n} + \sqrt{\frac{\bar{\gamma}}{n\gamma}} + \sqrt{\frac{H^2 L^2}{n\gamma}}(\log \gamma + \log n) \right\},$$

for some constant $\bar{c}_0 > 0$.

Note that the number of elements in $\mathfrak{I}(m)$ is upper bounded by $(m+1)^2$. Since $m$ is proportional to $n$, by setting $\bar{\gamma} = \bar{c}^* \log n$ for some constant $\bar{c}^* > 0$, we have $1 - 2(m+1)^2 \exp(-\bar{\gamma}) \to 1$. Consequently, it follows from Bonferroni's inequality that the following event occurs w.p.a.1 for any $\mathcal{I} \in \mathfrak{I}(m)$ such that $|\mathcal{I}| \geq c\gamma$,

$$\left| \frac{\sum_{i \in \mathbb{L}_\ell} \mathbb{I}(A_i \in \mathcal{I})\{Y_i - \mathrm{MLP}(X_i; \theta_{\mathcal{I}})\}^2}{|\mathbb{L}_\ell|\mathrm{Pr}(A \in \mathcal{I})} - \frac{\mathbb{E}\mathbb{I}(A \in \mathcal{I})\{Y - \mathrm{MLP}(X; \theta_{\mathcal{I}})\}^2}{\mathrm{Pr}(A \in \mathcal{I})} \right|$$

$$\leq O(1)\left\{ \frac{M \log n}{n} + \sqrt{\frac{\log n}{n\gamma}} + \sqrt{\frac{H^2 L^2}{n\gamma}} \log n \right\}, \tag{15}$$

where $O(1)$ denotes some positive constant. Under the given conditions, the RHS is $o(1)$. The proof is hence completed.

## D.2 PROOF OF LEMMA 2

Consider a given interval $\mathcal{I} \in \widehat{\mathcal{D}}^{(\ell)}$. We can find some interval $\mathcal{I}' \in \mathfrak{I}(m) \cap \widehat{\mathcal{D}}^{(\ell)}$ that is adjacent to $\mathcal{I}$. Consequently, the interval $\mathcal{I} \cup \mathcal{I}'$ belongs to $\mathfrak{I}(m)$ as well. It follows that

$$\frac{1}{|\mathbb{L}_\ell|}\sum_{i \in \mathbb{L}_\ell} \mathbb{I}(A_i \in \mathcal{I})\{Y_i - \widehat{q}_{\mathcal{I}}^{(\ell)}(X_i)\}^2 + \frac{1}{|\mathbb{L}_\ell|}\sum_{i \in \mathbb{L}_\ell} \mathbb{I}(A_i \in \mathcal{I}')\{Y_i - \widehat{q}_{\mathcal{I}'}^{(\ell)}(X_i)\}^2$$

$$\leq \frac{1}{|\mathbb{L}_\ell|}\sum_{i \in \mathbb{L}_\ell} \mathbb{I}(A_i \in \mathcal{I} \cup \mathcal{I}')\{Y_i - \widehat{q}_{\mathcal{I} \cup \mathcal{I}'}^{(\ell)}(X_i)\}^2 - \gamma. \tag{16}$$

By definition, $\widehat{q}_{\mathcal{I}\cup\mathcal{I}'}^{(\ell)}$ minimizes the loss $\sum_{i\in\mathbb{L}_i}\mathbb{I}(A_i\in\mathcal{I}\cup\mathcal{I}')\{Y_i-\widehat{q}_{\mathcal{I}\cup\mathcal{I}'}^{(\ell)}(X_i)\}^2$. It follows that

$$\sum_{i\in\mathbb{L}_\ell}\mathbb{I}(A_i\in\mathcal{I}\cup\mathcal{I}')\{Y_i-\widehat{q}_{\mathcal{I}\cup\mathcal{I}'}^{(\ell)}(X_i)\}^2\leq\sum_{i\in\mathbb{L}_\ell}\mathbb{I}(A_i\in\mathcal{I}\cup\mathcal{I}')\{Y_i-\widehat{q}_{\mathcal{I}'}^{(\ell)}(X_i)\}^2.$$

Combining this together with equation 16 yields that

$$\gamma\leq\frac{1}{|\mathbb{L}_\ell|}\sum_{i\in\mathbb{L}_\ell}\mathbb{I}(A_i\in\mathcal{I})\{Y_i-\widehat{q}_{\mathcal{I}'}^{(\ell)}(X_i)\}^2. \tag{17}$$

As both $Y$ and $\widehat{q}_{\mathcal{I}}^{(\ell)}$ are uniformly bounded, we have $\gamma\leq c|\mathbb{L}_\ell|^{-1}\sum_{i\in\mathbb{L}_\ell}\mathbb{I}(A_i\in\mathcal{I})$ for some constant $c>0$. By Bernstein's inequality, we have for any $t>0$ such that

$$\frac{1}{|\mathbb{L}_\ell|}\sum_{i\in\mathbb{L}_\ell}\mathbb{I}(A_i\in\mathcal{I})\leq\Pr(A\in\mathcal{I})+t, \tag{18}$$

with probability $1-\exp(-nt^2/\bar{c}(|\mathcal{I}|+t))$ for some constant $\bar{c}>0$.

Similar to equation 15, by setting $t_{\mathcal{I}}=\bar{c}_0\max(\sqrt{|\mathcal{I}|n^{-1}\log n},n^{-1}\log n)$ for some constant $\bar{c}_0>0$, we obtain w.p.a.1 that

$$\frac{1}{|\mathbb{L}_\ell|}\sum_{i\in\mathbb{L}_\ell}\mathbb{I}(A_i\in\mathcal{I})\leq\Pr(A\in\mathcal{I})+t_{\mathcal{I}},$$

for any $\mathcal{I}\in\mathfrak{I}(m)$. Under the condition that $b$ is continuous and $\mathcal{A}\times\mathcal{X}$ is compact, $b$ is bounded. Consequently, the probability $\Pr(A\in\mathcal{I})$ is proportional to the length of $\mathcal{I}$. In view of equation 18, for any $\mathcal{I}\in\widehat{\mathcal{D}}^{(\ell)}$, it shall satisfy

$$\frac{\gamma}{\bar{c}_1}\leq|\mathcal{I}|+\frac{\log n}{n}+2\sqrt{\frac{|\mathcal{I}|\log n}{n}}\leq 2|\mathcal{I}|+2\frac{\log n}{n},$$

for some constant $\bar{c}_1>0$, w.p.a.1, where the last inequality follows from Cauchy-Schwarz inequality. As $\gamma\gg n^{-1}\log n$, we obtain $|\mathcal{I}|\geq\gamma/\bar{c}_2$ for any $\mathcal{I}\in\widehat{\mathcal{D}}^{(\ell)}$ and some constant $\bar{c}_2>0$, w.p.a.1. The proof is hence completed.

### D.3 PROOF OF THEOREM 1

We begin with an outline of the proof. The proof is divide into four steps. In the first step, we show

$$\Pr(|\widehat{\mathcal{D}}^{(\ell)}|\leq|\mathcal{D}_\ell|)\to 1. \tag{19}$$

In the second step, we show

$$\max_{\tau\in J(\mathcal{D}_0)}\min_{\widehat{\tau}\in J(\widehat{\mathcal{D}}^{(\ell)})}|\widehat{\tau}-\tau|<\delta_{\min}, \tag{20}$$

where $\delta_{\min}=\min_{\mathcal{I}\in\mathcal{D}_0}|\mathcal{I}|/3$. By the definition of $\delta_{\min}$, this implies that

$$\Pr(|\widehat{\mathcal{D}}^{(\ell)}|\geq|\mathcal{D}_\ell|)\to 1. \tag{21}$$

Combining equation 22 together with equation 19 proves (i) in Theorem 1. This proves (ii) in Theorem 1. In the third step, we show

$$\max_{\tau\in J(\mathcal{D}_0)}\min_{\widehat{\tau}\in J(\widehat{\mathcal{D}}^{(\ell)})}|\widehat{\tau}-\tau|=o_p(1). \tag{22}$$

In the last step, we show (iii) holds. The proof is thus completed.

We next detail the proof for each of the step.

**Step 1.** Assume $|\mathcal{D}_0|>1$. Otherwise, equation 19 automatically holds. Consider the partition $\mathcal{D}=\{[0,1]\}$ which consists of a single interval and a zero q-function $q_{[0,1]}(x)=0$ for any $x$. By definition, we have

$$\sum_{\mathcal{I}\in\widehat{\mathcal{D}}^{(\ell)}}\sum_{i\in\mathbb{L}_\ell}\mathbb{I}(A_i\in\mathcal{I})\{Y_i-\widehat{q}_{\mathcal{I}}(X_i)\}^2+|\mathbb{L}_\ell|\gamma|\widehat{\mathcal{D}}^{(\ell)}|\leq\sum_{i\in\mathbb{L}_\ell}Y_i^2+|\mathbb{L}_\ell|\gamma.$$

Since $Y$ is uniformly bounded and $\gamma = o(1)$, the right-hand-side (RHS) is $O(n)$. Consequently, we obtain

$$|\widehat{\mathcal{D}}^{(\ell)}| \leq c_0 \gamma^{-1}, \tag{23}$$

for some constant $c_0 > 0$.

Notice that

$$\sum_{\mathcal{I} \in \widehat{\mathcal{D}}^{(\ell)}} \sum_{i \in \mathbb{L}_\ell} \mathbb{I}(A_i \in \mathcal{I})\{Y_i - \widehat{q}_{\mathcal{I}}^{(\ell)}(X_i)\}^2 \geq \underbrace{\sum_{\mathcal{I} \in \widehat{\mathcal{D}}^{(\ell)}} \sum_{i \in \mathbb{L}_\ell} \mathbb{I}(A_i \in \mathcal{I})\{Y_i - q_{\mathcal{I}}(X_i)\}^2}_{\eta_1}$$

$$+ \underbrace{\sum_{\mathcal{I} \in \widehat{\mathcal{D}}^{(\ell)}} \sum_{i \in \mathbb{L}_\ell} \mathbb{I}(A_i \in \mathcal{I})\{\widehat{q}_{\mathcal{I}}^{(\ell)}(X_i) - q_{\mathcal{I}}(X_i)\}^2}_{\eta_2} \tag{24}$$

$$- 2 \underbrace{\sum_{\mathcal{I} \in \widehat{\mathcal{D}}^{(\ell)}} \left| \sum_{i \in \mathbb{L}_\ell} \mathbb{I}(A_i \in \mathcal{I})\{Y_i - q_{\mathcal{I}}(X_i)\}\{\widehat{q}_{\mathcal{I}}^{(\ell)}(X_i) - q_{\mathcal{I}}(X_i)\} \right|}_{\eta_3}.$$

We next show $\eta_2, \eta_3 = o_p(1)$.

Consider $\eta_2$ first. Under Lemma 2, we have w.p.a.1 that

$$\eta_2 = \sum_{\mathcal{I} \in \widehat{\mathcal{D}}^{(\ell)}, |\mathcal{I}| \geq C\gamma} \sum_{i \in \mathbb{L}_\ell} \mathbb{I}(A_i \in \mathcal{I})\{\widehat{q}_{\mathcal{I}}^{(\ell)}(X_i) - q_{\mathcal{I}}(X_i)\}^2.$$

We decompose the RHS by

$$\sum_{\mathcal{I} \in \widehat{\mathcal{D}}^{(\ell)}, |\mathcal{I}| \geq C\gamma} |\mathbb{L}_\ell| \mathbb{E}[\mathbb{I}(A \in \mathcal{I})\{\widehat{q}_{\mathcal{I}}^{(\ell)}(X) - q_{\mathcal{I}}(X)\}^2 | \{O_i\}_{i \in \mathbb{L}_i}]$$

$$+ \sum_{\mathcal{I} \in \widehat{\mathcal{D}}^{(\ell)}, |\mathcal{I}| \geq C\gamma} \sum_{i \in \mathbb{L}_\ell} [\mathbb{I}(A_i \in \mathcal{I})\{\widehat{q}_{\mathcal{I}}^{(\ell)}(X_i) - q_{\mathcal{I}}(X_i)\}^2 - \mathbb{E}\{\mathbb{I}(A \in \mathcal{I})\{\widehat{q}_{\mathcal{I}}^{(\ell)}(X) - q_{\mathcal{I}}(X)\}^2 | \{O_i\}_{i \in \mathbb{L}_i}\}].$$

The first line is $o_p(n)$, by Lemma 1. Using similar arguments in equation 15, we can show the second term is upper bounded by

$$\bar{c}|\mathbb{L}_\ell| \sum_{\mathcal{I} \in \widehat{\mathcal{D}}^{(\ell)}, |\mathcal{I}| \geq C\gamma} \left\{ \frac{M \log n}{n} + \sqrt{\frac{|\mathcal{I}| H^2 L^2 \log n}{n}} \right\},$$

for some constant $\bar{c} > 0$, w.p.a.1. In view of equation 23, the above expression can be further bounded by

$$\bar{c} n \left\{ \frac{c_0 M \log n}{\gamma n} + \sum_{\mathcal{I} \in \widehat{\mathcal{D}}^{(\ell)}} \sqrt{\frac{|\mathcal{I}| H^2 L^2 \log n}{n}} \right\} \leq O(1) n \left\{ \frac{\log n}{\gamma n} + \sqrt{\frac{H^2 L^2 \log n}{\gamma n}} \right\},$$

where $O(1)$ denotes some positive constant and the second inequality follows from the Cauchy-Schwarz inequality. This yields that $\eta_2 = o_p(n)$.

Using similar arguments, we can show that $\eta_3 = o_p(n)$. It follows that

$$\sum_{\mathcal{I} \in \widehat{\mathcal{D}}^{(\ell)}} \sum_{i \in \mathbb{L}_\ell} \mathbb{I}(A_i \in \mathcal{I})\{Y_i - \widehat{q}_{\mathcal{I}}^{(\ell)}(X_i)\}^2 \geq \eta_1 + o_p(n). \tag{25}$$

Notice that

$$\eta_1 = \sum_{\mathcal{I}\in\widehat{\mathcal{D}}^{(\ell)}} \sum_{i\in\mathbb{L}_\ell} \mathbb{I}(A_i \in \mathcal{I})\{Y_i - Q(A_i, X_i) + Q(A_i, X_i) - q_{\mathcal{I}}(X_i)\}^2$$

$$= \underbrace{\sum_{\mathcal{I}\in\widehat{\mathcal{D}}^{(\ell)}} \sum_{i\in\mathbb{L}_\ell} \mathbb{I}(A_i \in \mathcal{I})\{Y_i - Q(A_i, X_i)\}^2}_{\eta_4} + \underbrace{\sum_{\mathcal{I}\in\widehat{\mathcal{D}}^{(\ell)}} \sum_{i\in\mathbb{L}_\ell} \mathbb{I}(A_i \in \mathcal{I})\{Q(A_i, X_i) - q_{\mathcal{I}}(X_i)\}^2}_{\eta_5}$$

$$+ \underbrace{2 \sum_{\mathcal{I}\in\widehat{\mathcal{D}}^{(\ell)}} \sum_{i\in\mathbb{L}_\ell} \mathbb{I}(A_i \in \mathcal{I})\{Y_i - Q(A_i, X_i)\}\{Q(A_i, X_i) - q_{\mathcal{I}}(X_i)\}}_{\eta_6}.$$

Using similar arguments in bounding $\eta_2$ and $\eta_3$, we can show $\eta_6 = o_p(n)$ and that

$$\eta_5 = |\mathbb{L}_\ell| \sum_{\mathcal{I}\in\widehat{\mathcal{D}}^{(\ell)}} \int_{\mathcal{I}} \mathbb{E}b(a|X)|Q(X, a) - q_{\mathcal{I}}(X)|^2 da + o_p(n).$$

The first term on the RHS is greater than $c_0 n \sum_{\mathcal{I}\in\widehat{\mathcal{D}}^{(\ell)}} \int_{\mathcal{I}} \mathbb{E}|Q(X, a) - q_{\mathcal{I}}(X)|^2 da$ for some constant $c_0 > 0$. It follows that

$$\eta_1 \geq \eta_4 + c_0 |\mathbb{L}_\ell| \sum_{\mathcal{I}\in\widehat{\mathcal{D}}^{(\ell)}} \int_{\mathcal{I}} \mathbb{E}|Q(X, a) - q_{\mathcal{I}}(X)|^2 da + o_p(n).$$

Note that under the step model assumption, $\eta_4$ can be rewritten as $\sum_{\mathcal{I}\in\mathcal{D}_0} \sum_{i\in\mathbb{L}_\ell} \mathbb{I}(A_i \in \mathcal{I})\{Y_i - q_{\mathcal{I}}(X_i)\}^2$. This together with equation 25, yields that

$$\sum_{\mathcal{I}\in\widehat{\mathcal{D}}^{(\ell)}} \sum_{i\in\mathbb{L}_\ell} \mathbb{I}(A_i \in \mathcal{I})(Y_i - \widehat{q}_{\mathcal{I}}^{(\ell)}(X_i))^2$$

$$\geq \sum_{\mathcal{I}\in\mathcal{D}_0} \sum_{i\in\mathbb{L}_\ell} \mathbb{I}(A_i \in \mathcal{I})(Y_i - q_{\mathcal{I}}(X_i))^2 + c_0|\mathbb{L}_\ell| \sum_{\mathcal{I}\in\widehat{\mathcal{D}}^{(\ell)}} \int_{\mathcal{I}} \mathbb{E}|Q(X, a) - q_{\mathcal{I}}(X)|^2 da + o_p(n),$$

For any integer $k$ such that $1 \leq k \leq |\mathcal{D}_0| - 1$, let $\tau_{0,k}^*$ be the change point location that satisfies $\tau_{0,k}^* = i/m$ for some integer $i$ and that $|\tau_{0,k} - \tau_{0,k}^*| < m^{-1}$. Denoted by $\mathcal{D}^*$ the oracle partition formed by the change point locations $\{\tau_{0,k}^*\}_{k=1}^{|\mathcal{D}_0|-1}$. Set $\tau_{0,0}^* = 0$, $\tau_{0,|\mathcal{D}_0|}^* = 1$ and $q_{[\tau_{0,k-1}^*, \tau_{0,k}^*)}^* = q_{[\tau_{0,k-1}, \tau_{0,k})}$ for $1 \leq k \leq |\mathcal{D}_0| - 1$ and $q_{[\tau_{0,K-1}^*, 1]}^* = q_{[\tau_{0,K-1}, 1]}$. Let $\Delta_k = [\tau_{0,k-1}^*, \tau_{0,k}^*) \cap [\tau_{0,k-1}, \tau_{0,k})^c$ for $1 \leq k \leq |\mathcal{D}_0| - 1$ and $\Delta_{|\mathcal{D}_0|} = [\tau_{0,|\mathcal{D}_0|-1}^*, 1] \cap [\tau_{0,|\mathcal{D}_0|-1}, 1]^c$. The length of each interval $\Delta_k$ is at most $m^{-1}$. It follows that

$$\sum_{\mathcal{I}\in\mathcal{D}^*} \sum_{i\in\mathbb{L}_\ell} \mathbb{I}(A_i \in \mathcal{I})\{Y_i - q_{\mathcal{I}}^*(X_i)\}^2 - \sum_{\mathcal{I}\in\mathcal{P}_0} \sum_{i\in\mathbb{L}_\ell} \mathbb{I}(A_i \in \mathcal{I})\{Y_i - q_{\mathcal{I}}(X_i)\}^2$$

$$\leq \sum_{k=1}^{|\mathcal{D}_0|} \sum_{i\in\mathbb{L}_i} \mathbb{I}(A_i \in \Delta_k)\Big\{Y_i^2 + \sup_{\mathcal{I}\subseteq[0,1]} \sup_x q_{\mathcal{I}}^2(x)\Big\}. \tag{26}$$

Using similar arguments in bounding the RHS of equation 17 in the proof of Lemma 2, we can show the RHS of equation 26 is $o_p(n)$, as $m$ diverges to infinity. Combining this together with equation 25 yields

$$\sum_{\mathcal{I}\in\widehat{\mathcal{D}}^{(\ell)}} \sum_{i\in\mathbb{L}_\ell} \mathbb{I}(A_i \in \mathcal{I})\{Y_i - \widehat{q}_{\mathcal{I}}^{(\ell)}(X_i)\}^2 \geq \sum_{\mathcal{I}\in\mathcal{D}^*} \sum_{i\in\mathbb{L}_\ell} \mathbb{I}(A_i \in \mathcal{I})\{Y_i - q_{\mathcal{I}}^*(X_i)\}^2$$

$$+ c_0|\mathbb{L}_\ell| \sum_{\mathcal{I}\in\widehat{\mathcal{D}}^{(\ell)}} \int_{\mathcal{I}} \mathbb{E}|Q(X, a) - q_{\mathcal{I}}(X)|^2 da + o_p(n). \tag{27}$$

By definition, we have

$$\sum_{\mathcal{I}\in\widehat{\mathcal{D}}^{(\ell)}} \sum_{i\in\mathbb{L}_\ell} \mathbb{I}(A_i \in \mathcal{I})(Y_i - \widehat{q}_{\mathcal{I}}^{(\ell)}(X_i))^2 + |\mathbb{L}_\ell|\gamma|\widehat{\mathcal{D}}^{(\ell)}|$$

$$\leq \sum_{\mathcal{I}\in\mathcal{P}^*} \sum_{i\in\mathbb{L}_\ell} \mathbb{I}(A_i \in \mathcal{I})\{Y_i - q_{\mathcal{I}}^*(X_i)\}^2 + |\mathbb{L}_\ell|\gamma|\mathcal{D}^*|.$$

Since $|\mathcal{D}^*| = |\mathcal{D}_0|$, it follows from equation 27 that

$$\sum_{\mathcal{I} \in \widehat{\mathcal{D}}^{(\ell)}} \int_{\mathcal{I}} \mathbb{E}|Q(X, a) - q_{\mathcal{I}}(X)|^2 da \leq \gamma\{|\mathcal{D}_0| - |\widehat{\mathcal{D}}^{(\ell)}|\} + o_p(1). \tag{28}$$

Note that the left-hand-side (LHR) is non-negative. As $\gamma > 0$, we obtain that $|\widehat{\mathcal{D}}^{(\ell)}| \leq |\mathcal{D}_0|$, w.p.a.1. This completes the proof for the first step.

**Step 2.** It follows from equation 28 that

$$\sum_{\mathcal{I} \in \widehat{\mathcal{D}}^{(\ell)}} \int_{\mathcal{I}} \mathbb{E}|Q(X, a) - q_{\mathcal{I}}(X)|^2 da \leq \gamma\{|\mathcal{D}_0| - 1\} + o_p(1),$$

and hence

$$\sum_{\mathcal{I} \in \widehat{\mathcal{D}}^{(\ell)}} \int_{\mathcal{I}} \mathbb{E}|Q(X, a) - q_{\mathcal{I}}(X)|^2 da \leq 2\gamma\{|\mathcal{D}_0| - 1\}, \tag{29}$$

w.p.a.1. We aim to show equation 20 holds under the event defined in equation 29. Otherwise, there exists some $\tau_0 \in J(\mathcal{P}_0)$ such that $|\hat{\tau} - \tau_0| \geq \delta_{\min}$, for all $\hat{\tau} \in J(\widehat{P})$. Under the event defined in equation 29, we obtain that

$$\int_{\tau_0 - \delta_{\min}}^{\tau_0 + \delta_{\min}} \mathbb{E}|Q(X, a) - q_{\mathcal{I}}(X)|^2 da \leq 2\gamma(|\mathcal{D}_0| - 1), \tag{30}$$

w.p.a.1. On the other hand, since $Q(X, a)$ is a constant function on $[\tau_0 - \delta_{\min}, \tau_0)$ or $[\tau_0, \tau_0 + \delta_{\min})$, we have

$$\int_{\tau_0 - \delta_{\min}}^{\tau_0 + \delta_{\min}} \mathbb{E}|Q(X, a) - q_{\mathcal{I}}(X)|^2 da$$
$$\geq \min_q \left(\delta_{\min}\mathbb{E}|q_{[\tau_0 - \delta_{\min}, \tau_0)}(X) - q(X)|^2 + \delta_{\min}\mathbb{E}|q_{[\tau_0, \tau_0 + \delta_{\min})}(X) - q(X)|_2^2\right)$$
$$\geq \frac{\delta_{\min}}{2}\mathbb{E}|q_{[\tau_0 - \delta_{\min}, \tau_0)}(X) - q_{[\tau_0, \tau_0 + \delta_{\min})}(X)|^2 \geq \frac{\delta_{\min}\kappa_0}{2},$$

where

$$\kappa_0 \equiv \min_{\substack{\mathcal{I}_1, \mathcal{I}_2 \in \mathcal{P}_0 \\ \mathcal{I}_1 \text{ and } \mathcal{I}_2 \text{ are adjacent}}} \mathbb{E}|q_{\mathcal{I}_1}(X) - q_{\mathcal{I}_2}(X)|^2 2 > 0.$$

This apparently violates equation 30 where $\gamma \leq \delta_{\min}\kappa_0/(4|\mathcal{D}_0 - 4|)$. equation 20 thus holds w.p.a.1.

**Step 3:** Combining the results obtained in the first two steps yields that $|\mathcal{D}^{(\ell)}| = |\mathcal{D}_0|$, w.p.a.1. It follows from equation 28 that

$$\sum_{\mathcal{I} \in \widehat{\mathcal{D}}^{(\ell)}} \int_{\mathcal{I}} \mathbb{E}|Q(X, a) - q_{\mathcal{I}}(X)|^2 da = o_p(1).$$

Using similar arguments in Step 2, we can show $\max_{\tau \in J(\mathcal{D}_0)} \min_{\hat{\tau} \in J(\widehat{\mathcal{D}}^{(\ell)})} |\hat{\tau} - \tau| < \epsilon$ w.p.a.1, for any sufficiently small $\epsilon > 0$. The proof is hence completed.

**Step 4:** We begin with some notations. For any $\pi$, we define a random policy $\pi_{\widehat{\mathcal{D}}^{(\ell)}}$ according to the partition $\widehat{\mathcal{D}}^{(\ell)}$ as follows:

$$\pi_{\widehat{\mathcal{D}}^{(\ell)}}(a|x) = \sum_{\mathcal{I} \subseteq \widehat{\mathcal{D}}^{(\ell)}} \mathbb{I}\{\pi(x) \in \mathcal{I}, a \in \mathcal{I}\}\frac{b(a|x)}{b(\mathcal{I}|x)}.$$

Note that $\int_0^1 \pi_{\widehat{\mathcal{D}}^{(\ell)}}(a|x)da = \sum_{\mathcal{I} \subseteq \widehat{\mathcal{D}}^{(\ell)}} \mathbb{I}\{\pi(x) \in \mathcal{I}\} = 1$ for any $x$. Consequently, $\pi_{\widehat{\mathcal{D}}^{(\ell)}}$ is a "valid" random policy.

The proposed value estimator $\widehat{V}^{(\ell)}$ is doubly-robust to $V(\pi_{\widehat{\mathcal{D}}^{(\ell)}})$. By Lemma 1, the estimated Q-function is consistent. Consequently, $\widehat{V}^{(\ell)}$ is consistent to $V(\pi_{\widehat{\mathcal{D}}^{(\ell)}})$. Since the propose estimator is a weighted average of $\widehat{V}^{(\ell)}$, it suffices to show $V(\pi_{\widehat{\mathcal{D}}^{(\ell)}})$ is consistent to $V(\pi)$. Note that

$$V(\pi_{\widehat{\mathcal{D}}^{(\ell)}}) = \mathbb{E} \int_{[0,1]} Q(X, a) \sum_{\mathcal{I} \subseteq \widehat{\mathcal{D}}^{(\ell)}} \mathbb{I}\{\pi(X) \in \mathcal{I}, a \in \mathcal{I}\} \frac{b(a|X)}{b(\mathcal{I}|X)} da$$

$$= \sum_{\mathcal{I}_0 \in \mathcal{D}_0} \int_{\mathcal{I}_0} \mathbb{E} q_{\mathcal{I}_0}(X) \sum_{\mathcal{I} \subseteq \widehat{\mathcal{D}}^{(\ell)}} \mathbb{I}\{\pi(X) \in \mathcal{I}\} \frac{b(\mathcal{I} \cap \mathcal{I}_0 | X)}{b(\mathcal{I}|X)}.$$

Similarly, we can show

$$V(\pi) = \sum_{\mathcal{I}_0 \in \mathcal{D}_0} \int_{\mathcal{I}_0} \mathbb{E} q_{\mathcal{I}_0}(X) \mathbb{I}\{\pi(X) \in \mathcal{I}_0\}. \tag{31}$$

For each $\mathcal{I}_0 \in \mathcal{D}_0$, there exists an interval $\mathcal{I} \in \widehat{\mathcal{D}}^{(\ell)}$ such that $|\mathcal{I} \cup \mathcal{I}_0|/|\mathcal{I}| \to 1$. We use $\widehat{\mathcal{I}}_0^{(\ell)}$ to denote this interval. Under the given conditions, we have

$$V(\pi_{\widehat{\mathcal{D}}^{(\ell)}}) = \sum_{\mathcal{I}_0 \in \mathcal{D}_0} \int_{\mathcal{I}_0} \mathbb{E} q_{\mathcal{I}_0}(X) \mathbb{I}\{\pi(X) \in \widehat{\mathcal{I}}_0^{(\ell)}\} + o_p(1).$$

In view of equation 31, the value difference $|V(\pi_{\widehat{\mathcal{D}}^{(\ell)}}) - V(\pi)|$ can be upper bounded by

$$\sum_{\mathcal{I}_0 \in \mathcal{D}_0} \Pr(\pi(X) \in \mathcal{I}_0 - \widehat{\mathcal{I}}_0^{(\ell)}) + \sum_{\mathcal{I}_0 \in \mathcal{D}_0} \Pr(\pi(X) \in \widehat{\mathcal{I}}_0^{(\ell)} - \mathcal{I}_0) + o_p(1).$$

The first two terms decay to zero under the given conditions on $\pi(X)$. The proof is hence completed.

### D.4 PROOF OF THEOREM 2

We need the following lemma to prove Theorem 2.

**Lemma 3** *Assume conditions in Theorem 2 hold. Then for any interval $\mathcal{I} \in \mathfrak{I}(m)$ with $|\mathcal{I}| \gg \gamma$ and any interval $\mathcal{I}' \in \widehat{\mathcal{D}}^{(\ell)}$ with $\mathcal{I} \subseteq \mathcal{I}'$, we have w.p.a.1 that*

$$\mathbb{E}|q_{\mathcal{I}}(X) - q_{\mathcal{I}'}(X)|^2 = o(1),$$

*where the little-o term is uniform in $\mathcal{I}$ and $\mathcal{I}'$.*

The rest of the proof is divided into three steps. In the first step, we show Assertion (i) in Theorem 2 holds. In the second step, we show Assertion (ii) in Theorem 2 holds. Finally, we present the proof for Lemma 3.

**Step 1.** Consider a sequence $\{d_n\}_n$ such that $d_n \to 0$ and $d_n \gg \gamma$. We aim to show $\inf_{\substack{a \in \mathcal{I}' \\ \mathcal{I}' \in \widehat{\mathcal{D}}^{(\ell)}}} \mathbb{E}[|Q(X, a) - \widehat{q}_{\mathcal{I}'}(X)|^2 | \{O_i\}_{i \in \mathbb{L}_\ell}] = o_p(1)$. By Lemma 1, it suffices to show $\inf_{\substack{a \in \mathcal{I}' \\ \mathcal{I}' \in \widehat{\mathcal{D}}^{(\ell)}}} \mathbb{E}|Q(X, a) - q_{\mathcal{I}'}(X)|^2 = o(1)$.

Suppose $|\mathcal{I}'| \geq d_n$. Then according to Lemma 3, we can find some $\mathcal{I}$ such that $a \in \mathcal{I} \subseteq \mathcal{I}', |\mathcal{I}| \to 0$, $\mathbb{E}|q_{\mathcal{I}}(X) - q_{\mathcal{I}'}(X)|^2 = o(1)$. Since $|\mathcal{I}| \to 0$ and $a \in \mathcal{I}$, it follows from the uniform continuity of the Q-function that $|q_{\mathcal{I}}(X) - Q(X, a)| \to 0$. The assertion thus follows.

Next, suppose $|\mathcal{I}'| < d_n$. Then $|\mathcal{I}'| \to 0$ as well. It follows from the uniform continuity of the Q-function that $\inf_{a \in \mathcal{I}'} |q_{\mathcal{I}'}(X) - Q(X, a)| \to 0$. The assertion thus follows. This completes the proof for the first step.

**Step 2.** Using similar arguments in Step 4 of the proof of Theorem 1, it suffices to show $V(\pi_{\widehat{\mathcal{D}}^{(\ell)}}) = V(\pi) + o_p(1)$. By definition,

$$V(\pi_{\widehat{\mathcal{D}}^{(\ell)}}) - V(\pi) = \sum_{\mathcal{I} \in \widehat{\mathcal{D}}^{(\ell)}} \int_{\mathcal{I}} \mathbb{E} Q(X, a) \mathbb{I}(\pi(X) \in \mathcal{I}) \frac{b(a|X)}{b(\mathcal{I}|X)} da - \mathbb{E} Q(\pi(X), X)$$

$$= \sum_{\mathcal{I} \in \widehat{\mathcal{D}}^{(\ell)}} \int_{\mathcal{I}} \mathbb{E}\{Q(X, a) - Q(\pi(X), X)\} \mathbb{I}(\pi(X) \in \mathcal{I}) \frac{b(a|X)}{b(\mathcal{I}|X)} da.$$

It follows that

$$|V(\pi_{\widehat{\mathcal{D}}^{(\ell)}}) - V(\pi)| \leq \sum_{\mathcal{I} \in \widehat{\mathcal{D}}^{(\ell)}} \inf_{a' \in \mathcal{I}} \int_{\mathcal{I}} \mathbb{E}|Q(a', X) - Q(\pi(X), X)|\mathbb{I}(\pi(X) \in \mathcal{I}) \frac{b(a|X)}{b(\mathcal{I}|X)} da$$

$$= \sum_{\mathcal{I} \in \widehat{\mathcal{D}}^{(\ell)}} \inf_{a' \in \mathcal{I}} \mathbb{E}|Q(a', X) - Q(\pi(X), X)|\mathbb{I}(\pi(X) \in \mathcal{I})$$

$$\leq \inf_{a', a'' \in \mathcal{I}, \mathcal{I} \in \widehat{\mathcal{D}}^{(\ell)}} \mathbb{E}|Q(a', X) - Q(a'', X)|.$$

In Step 1 of the proof, we have shown that $\inf_{a \in \mathcal{I}, \mathcal{I} \in \widehat{\mathcal{D}}^{(\ell)}} \mathbb{E}|Q(X, a) - q_{\mathcal{I}}(X)|^2 = o(1)$. It follows that $\inf_{a, a' \in \mathcal{I}, \mathcal{I} \in \widehat{\mathcal{D}}^{(\ell)}} \mathbb{E}|Q(X, a) - Q(a', X)|^2 = o(1)$ and hence $\inf_{a', a'' \in \mathcal{I}, \mathcal{I} \in \widehat{\mathcal{D}}^{(\ell)}} \mathbb{E}|Q(a', X) - Q(a'', X)| = o(1)$, by Cauchy-Schwarz inequality. This completes the proof for the second step.

**Step 3.** For a given interval $\mathcal{I}' \in \widehat{\mathcal{D}}^{(\ell)}$, the set of intervals $\mathcal{I}$ considered in Lemma 3 can be classified into the following three categories.

*Category 1:* $\mathcal{I} = \mathcal{I}'$. It is immediate to see that $q_{\mathcal{I}} = q_{\mathcal{I}'}$ and the assertion automatically holds.

*Category 2:* There exists another interval $\mathcal{I}^* \in \mathfrak{I}(m)$ that satisfies $\mathcal{I}' = \mathcal{I}^* \cup \mathcal{I}$. Notice that the partition $\widehat{\mathcal{D}}^{(\ell)*} = \widehat{\mathcal{D}}^{(\ell)} \cup \{\mathcal{I}^*\} \cup \mathcal{I} - \{\mathcal{I}'\}$ corresponds to another partition. By definition, we have

$$\frac{1}{|\mathbb{L}_\ell|} \sum_{i \in \mathbb{L}_\ell} \sum_{\mathcal{I}_0 \in \widehat{\mathcal{D}}^{(\ell)*}} \mathbb{I}(A_i \in \mathcal{I}_0)\{Y_i - \widehat{q}_{\mathcal{I}_0}(X_i)\}^2 + \gamma|\widehat{\mathcal{D}}^{(\ell)*}|$$

$$\geq \frac{1}{|\mathbb{L}_\ell|} \sum_{i \in \mathbb{L}_\ell} \sum_{\mathcal{I}_0 \in \widehat{\mathcal{D}}^{(\ell)}} \mathbb{I}(A_i \in \mathcal{I}_0)\{Y_i - \widehat{q}_{\mathcal{I}_0}(X_i)\}^2 + \gamma|\widehat{\mathcal{D}}^{(\ell)}|,$$

and hence

$$\frac{1}{|\mathbb{L}_\ell|} \sum_{i \in \mathbb{L}_\ell} \mathbb{I}(A_i \in \mathcal{I})\{Y_i - \widehat{q}_{\mathcal{I}}(X_i)\}^2 + \frac{1}{|\mathbb{L}_\ell|} \sum_{i \in \mathbb{L}_\ell} \mathbb{I}(A_i \in \mathcal{I}^*)\{Y_i - \widehat{q}_{\mathcal{I}^*}(X_i)\}^2$$

$$\geq \frac{1}{|\mathbb{L}_\ell|} \sum_{i \in \mathbb{L}_\ell} \mathbb{I}(A_i \in \mathcal{I}')\{Y_i - \widehat{q}_{\mathcal{I}'}(X_i)\}^2 - \gamma.$$

It follows from the definition of $\widehat{q}_{\mathcal{I}^*}$ that

$$\frac{1}{|\mathbb{L}_\ell|} \sum_{i \in \mathbb{L}_\ell} \mathbb{I}(A_i \in \mathcal{I}^*)\{Y_i - \widehat{q}_{\mathcal{I}^*}(X_i)\}^2 \leq \frac{1}{|\mathbb{L}_\ell|} \sum_{i \in \mathbb{L}_\ell} \mathbb{I}(A_i \in \mathcal{I}^*)\{Y_i - \widehat{q}_{\mathcal{I}'}(X_i)\}^2.$$

Therefore, we obtain

$$\frac{1}{|\mathbb{L}_\ell|} \sum_{i \in \mathbb{L}_\ell} \mathbb{I}(A_i \in \mathcal{I})\{Y_i - \widehat{q}_{\mathcal{I}}(X_i)\}^2 \geq \frac{1}{|\mathbb{L}_\ell|} \sum_{i \in \mathbb{L}_\ell} \mathbb{I}(A_i \in \mathcal{I})\{Y_i - \widehat{q}_{\mathcal{I}'}(X_i)\}^2 - \gamma. \quad (32)$$

*Category 3:* There exist two intervals $\mathcal{I}^*, \mathcal{I}^{**} \in \mathfrak{I}(m)$ that satisfy $\mathcal{I}' = \mathcal{I}^* \cup \mathcal{I} \cup \mathcal{I}^{**}$. Using similar arguments in proving equation 32, we can show that

$$\frac{1}{|\mathbb{L}_\ell|} \sum_{i \in \mathbb{L}_\ell} \mathbb{I}(A_i \in \mathcal{I})\{Y_i - \widehat{q}_{\mathcal{I}}(X_i)\}^2 \geq \frac{1}{|\mathbb{L}_\ell|} \sum_{i \in \mathbb{L}_\ell} \mathbb{I}(A_i \in \mathcal{I})\{Y_i - \widehat{q}_{\mathcal{I}'}(X_i)\}^2 - 2\gamma.$$

Hence, regardless of whether $\mathcal{I}$ belongs to Category 2, or it belongs to Category 3, we have

$$\frac{1}{|\mathbb{L}_\ell|} \sum_{i \in \mathbb{L}_\ell} \mathbb{I}(A_i \in \mathcal{I})\{Y_i - \widehat{q}_{\mathcal{I}}(X_i)\}^2 \geq \frac{1}{|\mathbb{L}_\ell|} \sum_{i \in \mathbb{L}_\ell} \mathbb{I}(A_i \in \mathcal{I})\{Y_i - \widehat{q}_{\mathcal{I}'}(X_i)\}^2 - 2\gamma. \quad (33)$$

Using similar arguments in equation 15, we can show w.p.a.1 that

$$\frac{1}{|\mathbb{L}_\ell|} \sum_{i \in \mathbb{L}_\ell} \mathbb{I}(A_i \in \mathcal{I})\{Y_i - \widehat{q}_{\mathcal{I}}(X_i)\}^2 = \mathbb{E}[\mathbb{I}(A \in \mathcal{I})\{Y - \widehat{q}_{\mathcal{I}}(X)\}^2|\{O_i\}_{i \in \mathbb{L}_\ell}] + o(\sqrt{\gamma|\mathcal{I}|}),$$

$$\frac{1}{|\mathbb{L}_\ell|} \sum_{i \in \mathbb{L}_\ell} \mathbb{I}(A_i \in \mathcal{I})\{Y_i - \widehat{q}_{\mathcal{I}'}(X_i)\}^2 = \mathbb{E}[\mathbb{I}(A \in \mathcal{I})\{Y - \widehat{q}_{\mathcal{I}'}(X)\}^2|\{O_i\}_{i \in \mathbb{L}_\ell}] + o(\sqrt{\gamma|\mathcal{I}|}),$$

where the little-$o$ terms are uniform in $\mathcal{I}$ and $\mathcal{I}'$. Combining these together with equation 33 yields

$$\mathbb{E}[\mathbb{I}(A \in \mathcal{I})\{Y - \widehat{q}_{\mathcal{I}}(X)\}^2|\{O_i\}_{i\in\mathbb{L}_\ell}] \geq \mathbb{E}[\mathbb{I}(A \in \mathcal{I})\{Y - \widehat{q}_{\mathcal{I}'}(X)\}^2|\{O_i\}_{i\in\mathbb{L}_\ell}] - 2\gamma + o(\sqrt{\gamma|\mathcal{I}|}),$$

for any $\mathcal{I}$ and $\mathcal{I}'$, w.p.a.1. Note that $q_{\mathcal{I}}$ satisfies $\mathbb{E}[\mathbb{I}(A \in \mathcal{I})\{Y - q_{\mathcal{I}}(X)\}|X] = 0$. We have

$$\mathbb{E}[\mathbb{I}(A \in \mathcal{I})\{q_{\mathcal{I}}(X) - \widehat{q}_{\mathcal{I}}(X)\}^2|\{O_i\}_{i\in\mathbb{L}_\ell}] \geq \mathbb{E}[\mathbb{I}(A \in \mathcal{I})\{q_{\mathcal{I}}(X) - \widehat{q}_{\mathcal{I}'}(X)\}^2|\{O_i\}_{i\in\mathbb{L}_\ell}] - 2\gamma + o(\sqrt{\gamma|\mathcal{I}|}).$$

Consider the first term on the RHS. Note that

$$\mathbb{E}[\mathbb{I}(A \in \mathcal{I})\{q_{\mathcal{I}}(X) - \widehat{q}_{\mathcal{I}'}(X)\}^2|\{O_i\}_{i\in\mathbb{L}_\ell}] = \mathbb{E}[\mathbb{I}(A \in \mathcal{I})\{q_{\mathcal{I}}(X) - q_{\mathcal{I}'}(X)\}^2|\{O_i\}_{i\in\mathbb{L}_\ell}]$$
$$+\mathbb{E}[\mathbb{I}(A \in \mathcal{I})\{\widehat{q}_{\mathcal{I}'}(X) - q_{\mathcal{I}'}(X)\}^2|\{O_i\}_{i\in\mathbb{L}_\ell}] - 2\mathbb{E}[\mathbb{I}(A \in \mathcal{I})\{q_{\mathcal{I}}(X) - q_{\mathcal{I}'}(X)\}\{\widehat{q}_{\mathcal{I}'}(X) - q_{\mathcal{I}'}(X)\}|\{O_i\}_{i\in\mathbb{L}_\ell}].$$

By Cauchy-Schwarz inequality, the last term on the RHS can be lower bounded by

$$-\frac{1}{2}\mathbb{E}[\mathbb{I}(A \in \mathcal{I})\{q_{\mathcal{I}}(X) - q_{\mathcal{I}'}(X)\}^2|\{O_i\}_{i\in\mathbb{L}_\ell}] - 2\mathbb{E}[\mathbb{I}(A \in \mathcal{I})\{\widehat{q}_{\mathcal{I}'}(X) - q_{\mathcal{I}'}(X)\}^2|\{O_i\}_{i\in\mathbb{L}_\ell}].$$

It follows that

$$\mathbb{E}[\mathbb{I}(A \in \mathcal{I})\{q_{\mathcal{I}}(X) - \widehat{q}_{\mathcal{I}'}(X)\}^2|\{O_i\}_{i\in\mathbb{L}_\ell}] \geq \frac{1}{2}\mathbb{E}[\mathbb{I}(A \in \mathcal{I})\{q_{\mathcal{I}}(X) - q_{\mathcal{I}'}(X)\}^2|\{O_i\}_{i\in\mathbb{L}_\ell}]$$
$$-3\mathbb{E}[\mathbb{I}(A \in \mathcal{I})\{\widehat{q}_{\mathcal{I}'}(X) - q_{\mathcal{I}'}(X)\}^2|\{O_i\}_{i\in\mathbb{L}_\ell}],$$

and hence

$$\frac{1}{2}\mathbb{E}[\mathbb{I}(A \in \mathcal{I})\{q_{\mathcal{I}}(X) - q_{\mathcal{I}'}(X)\}^2|\{O_i\}_{i\in\mathbb{L}_\ell}] - 2\gamma + o(\sqrt{\gamma|\mathcal{I}|}) \leq \mathbb{E}[\mathbb{I}(A \in \mathcal{I})\{q_{\mathcal{I}}(X) - \widehat{q}_{\mathcal{I}}(X)\}^2|\{O_i\}_{i\in\mathbb{L}_\ell}]$$
$$+3\mathbb{E}[\mathbb{I}(A \in \mathcal{I})\{q_{\mathcal{I}'}(X) - \widehat{q}_{\mathcal{I}'}(X)\}^2|\{O_i\}_{i\in\mathbb{L}_\ell}].$$

By Lemma 1 and the positivity assumption, the RHS is $o_p(|\mathcal{I}|)$. Note that the little-$o$ terms are uniform in $\mathcal{I}$ and $\mathcal{I}'$. As $|\mathcal{I}| \gg \gamma$, we obtain that

$$\mathbb{E}[\mathbb{I}(A \in \mathcal{I})\{q_{\mathcal{I}}(X) - q_{\mathcal{I}'}(X)\}^2|\{O_i\}_{i\in\mathbb{L}_\ell}] = o_p(|\mathcal{I}|),$$

uniformly for any $\mathcal{I}$ and $\mathcal{I}'$, or equivalently,

$$\mathbb{E}\left[\frac{b(\mathcal{I}|X)}{|\mathcal{I}|}\{q_{\mathcal{I}}(X) - q_{\mathcal{I}'}(X)\}^2|\{O_i\}_{i\in\mathbb{L}_\ell}\right] = o_p(1).$$

By the positivity assumption, we have

$$\mathbb{E}[\{q_{\mathcal{I}}(X) - q_{\mathcal{I}'}(X)\}^2|\{O_i\}_{i\in\mathbb{L}_\ell}] = o_p(1),$$

uniformly for any $\mathcal{I}$ and $\mathcal{I}'$, or equivalently,

$$\mathbb{E}\{q_{\mathcal{I}}(X) - q_{\mathcal{I}'}(X)\}^2 = o_p(1).$$

This yields that

$$\mathbb{E}\{q_{\mathcal{I}}(X) - q_{\mathcal{I}'}(X)\}^2 = o(1),$$

uniformly for any $\mathcal{I}$ and $\mathcal{I}'$. The proof is hence completed.

### D.5 PROOF OF THEOREM 3

In the first two steps, we compare the minimax convergence rate when $Q \in \mathcal{Q}_1$. In the next two steps, we compare the minimax convergence rate when $Q \in \mathcal{Q}_2$.

**Step 1:** We provide a lower bound for the minimax convergence of kernel-based OPE when $Q \in \mathcal{Q}_1$ in this step. Consider a piecewise constant Q-function

$$Q(a, x) = \begin{cases} 0, & \text{if } a \leq 1/2, \\ 1, & \text{otherwise.} \end{cases}$$

Apparently, we have $Q \in \mathcal{Q}_1$ when $C_2 \geq 1 \geq \epsilon$. Define a policy $\pi$ such that the density function of $\pi(X)$ equals

$$\begin{cases} 4/3, & \text{if } 1/4 \leq \pi(x) \leq 1/2, \\ 2/3, & \text{else if } 1/2 \leq \pi(x) < 4/3, \\ 0, & \text{otherwise.} \end{cases}$$

We aim to show for such $Q$ and $\pi$, the best possible convergence rate of kernel-based estimator is $n^{-1/3}$.

We first consider its variance. Since the conditional variance of $Y|A, X$ is uniformly bounded away from 0 and 1, similar to Theorem 1 of Colangelo & Lee (2020), we can show the variance is lower bounded by $O(1)(nh)^{-1}$ where $O(1)$ denotes some positive constant.

We next consider its bias. Since the behavior policy is know. The bias is equal to

$$\mathbb{E}\frac{K\{(A - \pi(X))/h\}}{hb(A|X)}\{Y - Q(\pi(X), X)\} = \mathbb{E}\frac{K\{(A - \pi(X))/h\}}{hb(A|X)}\{Q(A, X) - Q(\pi(X), X)\}$$

$$= \mathbb{E}\int_{\pi(X)-h/2}^{\pi(X)+h/2} K\left\{\frac{a - \pi(X)}{h}\right\}\{\mathbb{I}(\pi(X) \leq 1/2 < a) - \mathbb{I}(a \leq 1/2 < \pi(X))\}da.$$

Using the change of variable $a = ht + \pi(X)$, the bias equals

$$\mathbb{E}\int_{-1/2}^{1/2} K(t)\{\mathbb{I}(\pi(X) \leq 1/2 < \pi(X) + ht) - \mathbb{I}(\pi(X) + ht \leq 1/2 < \pi(X))\}dt.$$

Consider any $0 < h \leq \epsilon$ for some sufficiently small $\epsilon > 0$. The bias is then equal to

$$\frac{4}{3}\int_{1/2-\epsilon/2}^{1/2}\int_{-1/2}^{1/2} K(t)\{\mathbb{I}(a \leq 1/2 < a + ht) - \mathbb{I}(a + ht \leq 1/2 < a)\}dtda$$

$$+ \frac{2}{3}\int_{1/2}^{1/2+\epsilon/2}\int_{-1/2}^{1/2} K(t)\{\mathbb{I}(a \leq 1/2 < a + ht) - \mathbb{I}(a + ht \leq 1/2 < a)\}dtda.$$

Under the symmetric condition on the kernel function, the above quantity is equal to

$$\frac{2}{3}\int_{1/2-h/2}^{1/2}\int_{(1-2a)/2h}^{1/2} K(t)dtda \geq \frac{2}{3}\int_{1/2-h/2}^{1/2-h/4}\int_{(1-2a)/2h}^{1/2} K(t)dtda$$

$$\geq \frac{2}{3}\int_{1/2-h/2}^{1/2-h/4}\int_{1/4}^{1/2} K(t)dtda = \frac{h}{6}\int_{1/4}^{1/2} K(t)dt.$$

Consequently, the bias is lower bounded by $O(1)h$ where $O(1)$ denotes some positive constant.

To summarize, the root mean squared error of kernel based estimator is lower bounded by $O(1)\{(nh)^{-1/2} + h\}$ where $O(1)$ denotes some positive constant. The optimal choice of $h$ that minimizes such lower bound would be of the order $O(n^{-1/3})$. Consequently, the convergence rate is lower bounded by $O(1)n^{-1/3}$.

**Step 2:** We derive an upper bound for the minimax convergence rate of our estimator when $Q \in \mathcal{Q}_1$ in this step. Using similar arguments in the proof of Lemma 1 (see Section D.1) and the proof of Theorem 1 in Imaizumi & Fukumizu (2019), we can show that with proper choices of the MLP networks, the following holds with probability at least $1 - O(n^{-C})$ for some sufficiently large constant $C > 0$,

$$\max_{\mathcal{I}\in\mathfrak{I}(m), |\mathcal{I}|\geq c\gamma} \mathbb{E}[|q_{\mathcal{I}}(X) - \hat{q}_{\mathcal{I}}^{(\ell)}(X)|^2|\{O_i\}_{i\in\mathbb{L}_\ell}] \leq C(n|\mathcal{I}|)^{-2\beta/(2\beta+d)}\log^2 n, \tag{34}$$

where $\beta$ is defined in $\mathcal{Q}_1$ and $\mathcal{Q}_2$.

By equation 34 and the condition that $\gamma \gg n^{-2\beta/(2\beta+p)}\log^2 n$, using similar arguments in Steps 1 and 2 of the proof of Theorem 1, we can show that the estimated change point locations are consistent and $\mathcal{D}^{(\ell)} = \mathcal{D}_0$ with probability $1 - O(n^{-C})$. Similar to equation 27, it follows from equation 34 that

$$\sum_{\mathcal{I}\in\widehat{\mathcal{D}}^{(\ell)}}\int_{\mathcal{I}} \mathbb{E}|Q(X, a) - q_{\mathcal{I}}(X)|^2 da = O(n^{-2\beta/(2\beta+p)}),$$

up to some logarithmic factors. This further implies that the change point locations in $\widehat{D}^{(\ell)}$ converge at a rate of $O(n^{-2\beta/(2\beta+p)})$ up to some logarithmic factors.

We next establish the statistical properties of our value estimates. We first observe that when $b$ is known, our estimator is unbiased to the inverse-propensity score weighted estimator

$$\frac{1}{n}\sum_{\ell=1}^{\mathcal{L}}\sum_{\mathcal{I}\in\widehat{\mathcal{D}}^{(\ell)}}\sum_{i\in\mathbb{L}_\ell}\mathbb{I}(A_i\in\mathcal{I})\frac{\mathbb{I}(\pi(X_i)\in\mathcal{I})}{b(\mathcal{I}|X_i)}Y_i. \tag{35}$$

Consequently, its bias is equal to

$$\frac{1}{\mathcal{L}}\sum_{\ell=1}^{\mathcal{L}}\sum_{\mathcal{I}\in\widehat{\mathcal{D}}^{(\ell)}}\mathbb{E}\mathbb{I}(A\in\mathcal{I})\frac{\mathbb{I}(\pi(X)\in\mathcal{I})}{b(\mathcal{I}|X)}\{Q(A,X)-Q(\pi(X),X)\}$$

$$=\frac{1}{\mathcal{L}}\sum_{\ell=1}^{\mathcal{L}}\sum_{\mathcal{I}\in\widehat{\mathcal{D}}^{(\ell)}}\mathbb{E}\int_{a\in\mathcal{I}}\frac{\mathbb{I}(\pi(X)\in\mathcal{I})}{b(\mathcal{I}|X)}\{Q(a,X)-q_{\mathcal{I}}(X)\}b(a|X)da.$$

Since $b$ is bounded away from zero and infinity, and that $\sup_{\mathcal{I}}\Pr(\pi(X)\in\mathcal{I})/|\mathcal{I}|\le C_4$, the absolute value of the bias is upper bounded by

$$O(1)\frac{1}{\mathcal{L}}\sum_{\ell=1}^{\mathcal{L}}\sum_{\mathcal{I}\in\widehat{\mathcal{D}}^{(\ell)}}\mathbb{E}\int_{a\in\mathcal{I}}\frac{\mathbb{I}(\pi(X)\in\mathcal{I})}{|\mathcal{I}|}|Q(a,X)-q_{\mathcal{I}}(X)|da$$

$$\le O(1)\frac{1}{\mathcal{L}}\sum_{\ell=1}^{\mathcal{L}}\sum_{\mathcal{I}\in\widehat{\mathcal{D}}^{(\ell)}}\int_{a\in\mathcal{I}}\sup_x|Q(a,x)-q_{\mathcal{I}}(x)|da,$$

where $O(1)$ denotes some positive constant. For each $\mathcal{I}\in\widehat{\mathcal{D}}^{(\ell)}$, we could find some $\mathcal{I}_0\in\mathcal{D}_0$ such that the lengths of $\mathcal{I}-\mathcal{I}_0$ and $\mathcal{I}_0-\mathcal{I}$ are upper bounded by $O(n^{-2\beta/(2\beta+p)})$ up to some logarithmic factors. By definition, we have

$$q_{\mathcal{I}}(x)=\frac{\int_{a\in\mathcal{I}}\sum_{\mathcal{I}_0\in\mathcal{D}_0}b(a|x)\mathbb{I}(a\in\mathcal{I}_0)q_{\mathcal{I}_0}(x)da}{b(\mathcal{I}|x)}.$$

Consequently, $\sup_x|q_{\mathcal{I}}(x)-q_{\mathcal{I}_0}(x)|=O(n^{-2\beta/(2\beta+p)})$ up to some logarithmic factors. It follows that the absolute value of the bias of bias is upper bounded by $O(|\mathcal{D}^{(\ell)}|n^{-2\beta/(2\beta+p)})$ up to some logarithmic factors. As $|\mathcal{D}^{(\ell)}|=|\mathcal{D}_0|\le C_1$ with probability tending to 1, the absolute value of the bias is upper bounded by

$$O_p(n^{-2\beta/(2\beta+p)}), \tag{36}$$

up to some logarithmic factors.

We next consider the variance. The variance is of the same order of magnitude of that of equation 35. Since $\mathcal{L}$ is finite, it suffices to consider the variance of

$$\frac{\mathcal{L}}{n}\sum_{\mathcal{I}\in\widehat{\mathcal{D}}^{(\ell)}}\sum_{i\in\mathcal{L}_\ell}\mathbb{I}(A_i\in\mathcal{I})\frac{\mathbb{I}(\pi(X_i)\in\mathcal{I})}{b(\mathcal{I}|X_i)}Y_i. \tag{37}$$

The variance of equation 37 is upper bounded by

$$O(n^{-1})\mathbb{E}\left\{\sum_{\mathcal{I}\in\widehat{\mathcal{D}}^{(\ell)}}\mathbb{I}(A\in\mathcal{I})\frac{\mathbb{I}(\pi(X)\in\mathcal{I})}{b(\mathcal{I}|X)}\right\}^2=O(n^{-1})\left\{\sum_{\mathcal{I}\in\widehat{\mathcal{D}}^{(\ell)}}\mathbb{E}\mathbb{I}(A\in\mathcal{I})\frac{\mathbb{I}(\pi(X)\in\mathcal{I})}{b^2(\mathcal{I}|X)}\right\}$$

$$=O(n^{-1})\left\{\sum_{\mathcal{I}\in\widehat{\mathcal{D}}^{(\ell)}}\mathbb{E}\frac{\mathbb{I}(\pi(X)\in\mathcal{I})}{b(\mathcal{I}|X)}\right\}=O(n^{-1}|\widehat{\mathcal{D}}^{(\ell)}|),$$

where the last equality is due to that $b$ is uniformly bounded way from zero and that $\Pr(\pi(X)\in\mathcal{I})\le O(1)|\mathcal{I}|$ for some positive constant $O(1)$. As $|\mathcal{D}^{(\ell)}|=|\mathcal{D}_0|\le C_1$ with probability tending to 1, the variance of equation 37 is upper bounded by $O_p(n^{-1})$. Consequently, the variance of our value estimator is bounded by $O_p(n^{-1})$ as well. In addition, we note this bound and the bias bound

in equation 36 is uniform in $Q \in \mathcal{Q}_1 \cup \mathcal{Q}_2$ and $\pi \in \Pi$. The proposed value converges at a rate of $O_p(n^{-2\beta/(2\beta+p)})$ up to some logarithmic terms. When $2\beta/(2\beta+p) > 1/2$, both the bias and the variance decay at a rate of $n^{-1/2}$ up to some logarithmic factors. This completes the proof for the first part.

**Step 3.** We provide a lower bound for the minimax convergence of kernel-based OPE when $Q \in \mathcal{Q}_2$ in this step. Similar to Step 1, we can show its variance is lower bounded by $O(n^{-1}h^{-1})$. When $h \asymp n^{-\kappa}$ for some $\kappa > 3/5$, the root mean squared error is larger than or equal to $O(n^{-(1-\kappa)/2})$.

Consider the Q-function

$$Q(x,a) = Ch^{-1}K\left(\frac{a - \pi(x)}{h}\right),$$

for some constant $C > 0$. With proper choice of $C$, we can show that such a choice of $Q$-function belongs to $\mathcal{Q}_2$. Using similar arguments in Step 1, we can show the bias equals

$$\mathbb{E}C^{-1}\frac{K^2\{(A - \pi(X))/h\}}{h^2 b(A|X)} \geq C^{-1}\mathbb{E}\frac{K^2\{(A - \pi(X))/h\}}{h^2}.$$

Using similar arguments in Step 1, we can show the right-hand-side is lower bounded $O(1)h$. When $h \asymp n^{-\kappa}$ for some $\kappa < 1/5$, the root mean squared error is larger than or equal to $O(n^{-\kappa})$.

To summarize, when $h \asymp n^{-\kappa}$ for some $\kappa < 1/5$ or $\kappa > 3/5$, kernel-based OPE converges at a rate of $n^{-\kappa^*}$ for some $\kappa^* < 1/5$.

**Step 4.** We provide an upper bound for the minimax convergence of our estimator when $Q \in \mathcal{Q}_2$ in this step.

Consider its bias first. Using similar arguments in Step 2, the bias is upper bounded by $O(\mathcal{L}^{-1})\sum_{\ell=1}^{\mathcal{L}}\sum_{\mathcal{I}' \in \widehat{\mathcal{D}}^{(\ell)}}\int_{a \in \mathcal{I}'}\sup_x |Q(a,x) - q_{\mathcal{I}'}(x)|da$. Similar to Lemma 3, we can show for any interval $\mathcal{I} \in \mathfrak{I}(m)$ with $|\mathcal{I}| \gg \gamma$ and any interval $\mathcal{I}' \in \widehat{\mathcal{D}}^{(\ell)}$ with $\mathcal{I} \subseteq \mathcal{I}'$, we have w.p.a.1 that

$$\mathbb{E}|q_{\mathcal{I}}(X) - q_{\mathcal{I}'}(X)|^2 = O(1)\sqrt{\frac{\gamma}{|\mathcal{I}|}}, \tag{38}$$

provided that $\gamma$ is proportional to $n^{-2\beta/(2\beta+p)}$ up to some logarithmic factors. For any $a$, there exists an interval $\mathcal{I}$ whose length is proportional to $\gamma \log n$ that covers $a$. Since $Q \in \mathcal{Q}_2$, we have $\sup_x |Q(a,x) - q_{\mathcal{I}}(x)| = O(|\mathcal{I}|)$. This together with equation 38 yields that

$$\sum_{\mathcal{I}' \in \widehat{\mathcal{D}}^{(\ell)}}\int_{a \in \mathcal{I}'}\sup_x |Q(a,x) - q_{\mathcal{I}'}(x)|da \leq O(1)\left(|\mathcal{I}| + \sqrt{\frac{\gamma}{|\mathcal{I}|}}\right).$$

Set $\mathcal{I} = \gamma^{1/3}$, the bias is upper bounded by $O(1)\gamma^{1/3}$. Using similar arguments in Step 2 of the proof, the standard deviation is upper bounded by $\sum_{\ell=1}^{\mathcal{L}}\mathcal{L}^{-1}\sqrt{n^{-1}|\widehat{\mathcal{D}}^{(\ell)}|}$. By equation 23, It is upper bounded by $O(n^{-1/2}\gamma^{-1/2})$. By setting $\gamma$ to be proportional to $n^{-3/5}$ (this rate is achievable under the condition that $4\beta > 3p$), we obtain the rate of $n^{-1/5}$.

