# OpenReview forum: "Deep Jump Q-Evaluation for Offline Policy Evaluation in Continuous Action Space"
_ICLR.cc/2021/Conference — Reject_

### Official Review · AnonReviewer1 · 2020-10-26
**Recommendation for "Deep Jump Q-Evaluation for Offline Policy Evaluation in Continuous Action Space"**

**Rating:** 8
**Confidence:** 1

**Review:**

Summary of paper:
The main contribution of this paper is a new algorithm to learn the expected reward function for a given target policy using the historical data generated by a different behavior policy in continuous action domains.  All current Offline-Policy Evaluation (OPE) methods for handling continuous action domains use a kernel function to extend Inverse Probability Weighting (IPW) or Doubly Robust (DR) approaches for discrete action domains.  The algorithm proposed in this work adaptively discretizes the action space by combining methods in multi-scale changepoint detection, multi-layer perceptron regression and OPE in discrete action domains.  The finite sample performance of the proposed method, known as Deep-Jump Q-Evaluation (DJQE), is compared to that of two kernel-based methods, one due to Kallus and Zhou (2018) and another due to Colangelo and Lee (2020), on synthetic as well as real-world data.  To generate synthetic data, four scenarios are considered, where in each case the Q-function is continuous in the action domain or is a piecewise function of the action.  In almost all of these cases, DJQE outperforms the two kernel-based methods.  Similarly, when applied to real-world Warfarin data (after calibration), DJQE outperforms the two kernel-based methods with respect to the bias, standard deviation and mean squared error, even when the sample size is small (n=50).  The average runtime of DJQE in each scenario (for synthetic or real-world data) is about 5 minutes.

Plus points:
- The experimental results seem to demonstrate quite convincingly that DJQE outperforms the two kernel-based methods in almost all cases.
- The methodology seems sound.
- The theoretical results also appear correct and prove the soundness of the method for a fairly wide range of functions - those that are continuous in the feature space and action domain, as well as those that are piecewise constant.
- The method can model jump discontinuities in the Q-function.

Questions:
- Why is it reasonable to assume that the Q-function can be well-approximated using piecewise linear combinations of MLPs?
- How is the performance of DJQE affected by the choice of the regularization parameter \gamma?

Minor comments/questions:

- Page 2, line -2: exists -> to exist
- Page 3, line 4 of Section 2.3: segments -> segment
- Page 3, line -4: Was there a reason for choosing the logarithm function here?
- Page 4, lines 12 to 13: Is there a theoretical justification for such a choice of m, or is it based on empirical observations?  Also, to what extent does the performance of DJQE depend on the initial choice of m?
- Page 5, Equation (4): Could it be justified why the minimizer is unique?
- Page 5, third line after Equation (4): Should it be Figure 3, Appendix A?  (There is no Figure A.)
- Page 5, line -4: How is \hat{Q} used in the solution of Equation (5)?
- Page 6, Assumption 1: "...number of nodes [in] each hidden layer..."
- Page 6, last line of the statement of Theorem 1: Should D be D_0?
- Page 6, lines -7 to -6: I did not fully understand what this means; do the change points of \hat{D}^{\ell} vary with m?
- Page 7, line -7: data -> dataset
- Page 7, line -5: How was the exponent 0.2 chosen?
- Page 7, last line, and Page 8, line 9 of Section 5.2: "...with 10 hidden [layers]..."
- Fifth reference on Page 10: Double occurrence of "Technical report".

* Update after reading author(s)' response: Thank you very much for the detailed answers to my questions (as well as the other reviewers' comments/questions).  I have upgraded my score; wishing you all the best.

---

### Official Review · AnonReviewer4 · 2020-10-28
**An efficient offline policy evaluation method for non-continuous Q functions**

**Rating:** 6
**Confidence:** 3

**Review:**

Summary

This paper proposes a new method for offline evaluation when the action space is continuous, one dimensional. This overcomes the drawbacks of the kernel based method, which cannot be applied to non-smooth Q functions and requires heavy computation to optimize the bandwidth. The proposed method can be applied to discontinuous Q functions like step functions, and achieves smaller bias. This is made possible by the adaptive jump q learning method.


Pros

While the kernel method requires a single bandwidth to control the bias and variance of the value estimator, the proposed method adapts to the shape of the Q function by dividing the action space in an adaptive way, so that the MLP fitted in each interval of the action space approximates well the real Q function. Hence, the intervals can have possibly different lengths according to the shape of the true Q function. A multi-scale change point detection method is used for determining the intervals, which requires only a linear computational cost.

Experiment results are convincing.

Cons

1) In Algorithm 1, “Collect cost function step” computes MLP regressor for every possible interval. Hence, computation will become heavy when the number of initial intervals (m) is large. Authors should add discussion about this point.

2) Some notations are confusing


Minor comments

1) Gamma appears before it is defined.
2) L is both the number of subsets and the numer of layers in neural networks.
Are they meant to be the same (as they increase with n), or are they different? In the latter case, they should be distinguished.

---

### Official Review · AnonReviewer2 · 2020-10-29
**A well written paper for continuous Q evaluation with both empirically and theoretically justified results.**

**Rating:** 6
**Confidence:** 3

**Review:**

This paper considers the problem of off-policy evaluation with continuous actions. The main idea is to first using multi-scale change point detection to discretize the action space and then apply traditional IPW or DR methods to estimate the value. The DJQE method is theoretically analyzed under both the cases that the Q function is either a piecewise function or a continuous function. For continuous function, it is not surprising that as the number of splits m goes to infinity as n, the estimation is consistent, while additional results in Theorem 2 also shows that for limited m, the estimator can also be shown as a uniform approximation of the Q value. Experiments consider both a toy dataset and a real problem in personalized does finding, and the results show that the DJQE method is superior than existing methods for continuous Q evaluation.

The paper is clearly written and easy to follow. I only have a few comments:

1. In the experiments, since computing the optimal bandwidth is very time consuming for the baseline methods, it would good to provide a detailed computation cost comparison.
2. As mentioned in the method part, m is initially set to be proportional to n, and the final partition size is much smaller than m. Would the authors shows these detailed numbers in the experiments?
3. It could be great if more real-world problems can be evaluated in the current experimental section, such as the dynamic pricing example introduced previously.

---

### Official Review · AnonReviewer3 · 2020-11-01
**review for Deep Jump Q-evaluation**

**Rating:** 5
**Confidence:** 3

**Review:**

Summary:

This paper proposes a new method that adaptively merges intervals to form a discrete action space where on each interval Q_I values are learned via deep neural networks. Then it applies ready-methods designed for discrete action spaces to do off-policy evaluation.

##########################################################################

Reasons for score:

The paper offers a new way to apply methods designed for discrete action spaces onto continuous action spaces and it seems to perform better than the two chosen baselines as seen from the experiment results. Although the authors mentioned problems with the baseline models quickly, it would be nice to see a more in-depth analysis in the experiments to demonstrate these problems that this paper has set out to overcome. It is also not very clear to me when and why DJQE performs better than the baselines (does it always perform better than the baselines?). I gave a conservative score 4 but I'm willing to change my evaluation if convinced.

##########################################################################

Pros:

The paper provided theoretical support to the proposed method by proving its consistency under two reasonable assumptions. The method was tested on both synthetic data and simulated real world data.

##########################################################################

Cons:

Overall the paper is not very clear to me. It would be nice to see more in-depth theoretical analysis on the main advantages of DJQE compared to the baselines, the lack of which generates the following questions:
 - Will this method always achieve lower biases than baselines on new datasets? I'm not sure about the quality of evaluation from a simulation model on the personalized dose finding application.
 - What are the potential problems/limitations of DJQE if there are any?
 (Although these questions are commonly raised on methods that rely on experimental proofs of their superior performance, they seem particularly relevant for this paper.)

##########################################################################

Questions during rebuttal period:

 - How does the computational cost of DJQE scale with a decreasing maximum threshold of bias?

 - How accurate is the simulation model trained on Warfarin?

---

### Decision · Program_Chairs · 2021-01-07
**Final Decision**

**Decision:**

Reject

**Comment:**

The paper considers the OPE problem under the contextual bandit model with continuous action.  They studied the model of a piecewise constant value function according to the actions.   The assumption is new, though still somewhat restrictive as it requires the piecewise constant partitions to be the same for all x.  The proposed algorithm estimates the partitions, and then used it to build a doubly robust estimator with stratified importance sampling (fitting an MLP for each partition separately).


The reviewers have mixed views about the paper.  The following is the AC's evaluation based on reading the paper and consolidating the reviewers' comments and the authors' responses.

Pros:

- The algorithm is new and it makes sense for the new problem setup  (though computationally intractable)
- The experimental results outperform the baseline and reinforces the theory. But it's a toy example at best.

Cons:

- The method is called "Q-learning" but it is somewhat disappointing to see that it actually applies only to the contextual bandit model (without dynamics).  There is quite a bit of branding issues here. I suggest the authors to revise it to reflect the actual problem setup.

-  The estimator is assumed to be arg min, but the objective function is non-convex and cannot be solved efficiently in general, e.g., (3) involves searching over all partitions... and (4) involves solving neural network partitions.  In other words, the result applies to a hypothetical minimizer that the practical solvers may or may not obtain (the authors cited Scikit-Learn for the optimization algorithm and claims that the optimization problem can be solved, which is not the case ...  the SGD algorithm can be applied to solve it, but it does not necessarily find you the solution).

- The theory is completely asymptotic and generic. There is no rate of convergence specified, and no dependence on the number of jumps |D_0| at all in Theorem 1.

-  Theorem 3 is obnoxiously sloppy. The assumptions are not made explicit (do you need Assumption 1 and 2, what is the choice of \rho? ) The notion of "minimax rate" is not defined at all.   Usually the minimax rate is the property of problem setting,  i.e., Min over all algorithms, and Max over all problems with in a family.  However, in the way the authors described the results in Theorem 3,  it says the "the minimax convergence rate of kernel-based estimator is Op(n^{−1/3})."  which seems to be restricting the algorithms instead.  Such non-typical choices require clear definitions and justification.    Based on what is stated, it really appears that the authors are just comparing upper bounds of the two methods.

I looked at the appendix and while there is a "lower bound analysis", the bound is not information-theoretical, but rather a fixed example where an unspecified family of algorithms (I think it is a specific kernel smoothing method with a arbitrary choice of the bandwidth parameter h) will fail.

Suggestions to the authors:

-  Instead of a piecewise constant (and uniformly bounded) function, why not consider the total variation class, which is strictly more general and comes with the same rate?

- For formalizing the lower bound, I suggest the authors to look into classical lower bounds for linear smoother, e.g., Donoho, Liu, MacGibbon (1990); which clearly illustrates that kernel smoothing-type methods do not achieve the minimax rates; and that wavelets-based approaches, locally adaptive regression splines, and fused lasso (You can think about the  Haar Wavelets as a basis function of piecewise linear functions ) do.

The authors can improve the paper by ensuring that the theoretical parts are clearly and rigorously presented; and perhaps to iron out the more useful finite-sample analysis that depends on model parameters of interest.